# The genomic landscape of human cellular circadian variation points to a novel role for the signalosome

**Ludmila Gaspar[1†], Cedric Howald[2,3], Konstantin Popadin[2‡], Bert Maier[4], Daniel Mauvoisin[5§], Ermanno Moriggi[1], Maria Gutierrez-Arcelus[2,3], Emilie Falconnet[2,3], Christelle Borel[2,3], Dieter Kunz[6], Achim Kramer[4], Frederic Gachon[5#], Emmanouil T Dermitzakis[2,3], Stylianos E Antonarakis[2,3], Steven A Brown[1]\***

[1]Institute of Pharmacology and Toxicology, University of Zurich, Zurich, Switzerland; [2]Department of Genetic Medicine and Development, University of Geneva, Geneva, Switzerland; [3]Institute of Genetics and Genomics in Geneva, University of Geneva, Geneva, Switzerland; [4]Charité–Universitätsmedizin, Laboratory of Chronobiology, Berlin, Germany; [5]Department of Pharmacology and Toxicology, University of Lausanne, Lausanne, Switzerland; [6]Institute of Physiology, Charité-Universitätsmedizin Berlin, Working Group Sleep Research & Clinical Chronobiology, Berlin, Germany

**\*For correspondence:** Steven. brown@pharma.uzh.ch

**Present address:** [†]Max-Planck Institute, Tuebingen, Germany; [‡]Immanuel Kant Baltic Federal University, Kaliningrad, Russia; [§]Institute of Bioengineering, School of Life Sciences, Ecole PolytechniqueFédérale de Lausanne and Swiss Institute of Bioinformatics, Lausanne, Switzerland; [#]Diabetes and Circadian Rhythms department, Nestlé Institute of HealthSciences, Lausanne, Switzerland

**Competing interests:** The authors declare that no competing interests exist.

**Abstract** The importance of natural gene expression variation for human behavior is undisputed, but its impact on circadian physiology remains mostly unexplored. Using umbilical cord fibroblasts, we have determined by genome-wide association how common genetic variation impacts upon cellular circadian function. Gene set enrichment points to differences in protein catabolism as one major source of clock variation in humans. The two most significant alleles regulated expression of COPS7B, a subunit of the COP9 signalosome. We further show that the signalosome complex is imported into the nucleus in timed fashion to stabilize the essential circadian protein BMAL1, a novel mechanism to oppose its proteasome-mediated degradation. Thus, circadian clock properties depend in part upon a genetically-encoded competition between stabilizing and destabilizing forces, and genetic alterations in these mechanisms provide one explanation for human chronotype.

DOI: https://doi.org/10.7554/eLife.24994.001

## Introduction

A biological 'circadian' clock governs nearly all aspects of human behavior and physiology in synchrony with the geophysical day. Although the basic mechanism of this clock is highly conserved across evolution, humans show widely varying phases of behavior (chronotypes) ranging from very early (so-called 'larks') to very late ('owls'). In rare instances of familial circadian disorders such as advanced sleep phase syndrome, it has been possible to identify specific mutations in dedicated 'clock proteins' that affect human circadian behavior and the period of the underlying clock (*Toh et al., 2001*; *Xu et al., 2005*). More recently, these investigations have been complemented by genome-wide association (GWAS) studies using questionnaire-based methods to interrogate genotyped cohorts about diurnal preference. These strategies have unearthed significant polymorphisms in regions containing known clock genes as well as other loci, but linking novel loci to biological

mechanisms remains challenging (*Allebrandt and Roenneberg, 2008*; *Hu et al., 2016*; *Jones et al., 2016*; *Lane et al., 2016*).

In cell- and tissue-based studies, it has been possible to link GWAS-based insights to biochemical mechanisms using eQTLs (expression quantitative trait loci) based upon transcript levels or other cellular parameters. Typically, because of the labor involved in collecting and analyzing cellular material, the cohorts used in such investigations are much smaller (hundreds of samples rather than >10,000 in conventional GWAS). Partially compensating for this deficiency is the relative simplicity of the analytical system and the precision of expression trait measurement (*Nica and Dermitzakis, 2013*). Such approaches have been used successfully in a variety of contexts, identifying susceptibility loci for autoimmune disease (*Kochi, 2016*), for HIV infection (*Loeuillet et al., 2008*), and multiple other diseases. Since individual common alleles typically have rather small effect sizes, eQTL analysis provides a cellular methodology to analyse disease-relevant molecular phenotypes in simple systems that can amplify their effects (*Hou and Zhao, 2013*).

Since circadian clock function is cell-autonomous, it would be ideally suited to eQTL-based investigations. The fundamental unit of circadian timekeeping is comprised of feedback loops of transcription and translation of dedicated 'clock genes', coupled to parallel and perhaps even independent cycles of post-translational modification. These highly conserved clocks are hierarchically organized in mammals including man, with a 'master' clock in the suprachiasmatic nuclei (SCN) of the hypothalamus driving diurnal behavior and physiology both directly and indirectly via 'slave' oscillators of similar mechanism elsewhere in the brain and body (*Brown and Azzi, 2013*).

In the recent past, circadian clocks in cultured fibroblast cells have been used as a workhorse for understanding circadian function. Importantly, interfering with these clocks produces clock phenotypes similar to those present in a corresponding whole-animal context (*Brown et al., 2005*). Questionnaires about human daily behavior are even predictive of fibroblast clock properties from the same individuals: in general, cells from 'larks' show a short period, and those from 'owls' a longer period (*Brown et al., 2008*). Herein, we have exploited these cellular proxies to explore the genetic basis of variation in human circadian clock function.

## Results

### eQTL GWAS identifies polymorphisms influencing cellular circadian function

To assay circadian clock function in cultured cells, differently-phased clocks in the cells of the culture dish are synchronized by rapid chemical induction of a clock gene, and then circadian gene expression is monitored for several days, typically via a synthetic reporter (*Gaspar and Brown, 2015*). With this aim, we transduced the 159 umbilical cord fibroblasts of the Genecord II library (*Borel et al., 2011*; *Dimas et al., 2009*) – each genotyped for 2.5 million common single-nucleotide polymorphisms, or SNPs – with a lentiviral *BMAL1-luciferase* circadian reporter, then synchronized clocks with dexamethasone, and measured acute induction of *PER1* and *PER2* gene expression via qPCR, as well as the period, phase, and amplitude of subsequent circadian oscillation via real-time in vitro bioluminescence recording. For each parameter, large inter-individual differences were observed (*Figure 1—figure supplement 1a,b*), as has been reported previously (*Brown et al., 2005*). Normalized data were used as quantitative traits for genome-wide association, identifying for each clock parameter polymorphisms at different confidence levels up to $p=10^{-7}$ (*Figure 1a*, see individual Manhattan and Q-Q plots in *Figure 1—figure supplement 1c–g*, *Figure 1—figure supplement 2*; all significant SNPs are listed in *Figure 1—source data 1*).

We next compared SNPs across the traits that we identified. Considering alleles achieving 'suggestive' genome-wide significance ($p<10^{-5}$), we see correlation between multiple pairs of traits, notably *PER1* and *PER2* expression (p=0.024), period and phase (p=1.48e-10), and period and amplitude (p=0.009) (*Supplementary file 1*). Given the close homology and overlapping function of the PER proteins, and the known correlation between period length and phase, these correlations are expected. Surprisingly, however, the most significant alleles in any one category are not among the most significant alleles in another, suggesting relatively independent genetic regulation of key circadian clock parameters ('state variables') for small changes, and the resilience of the overall circadian mechanism to small perturbations.

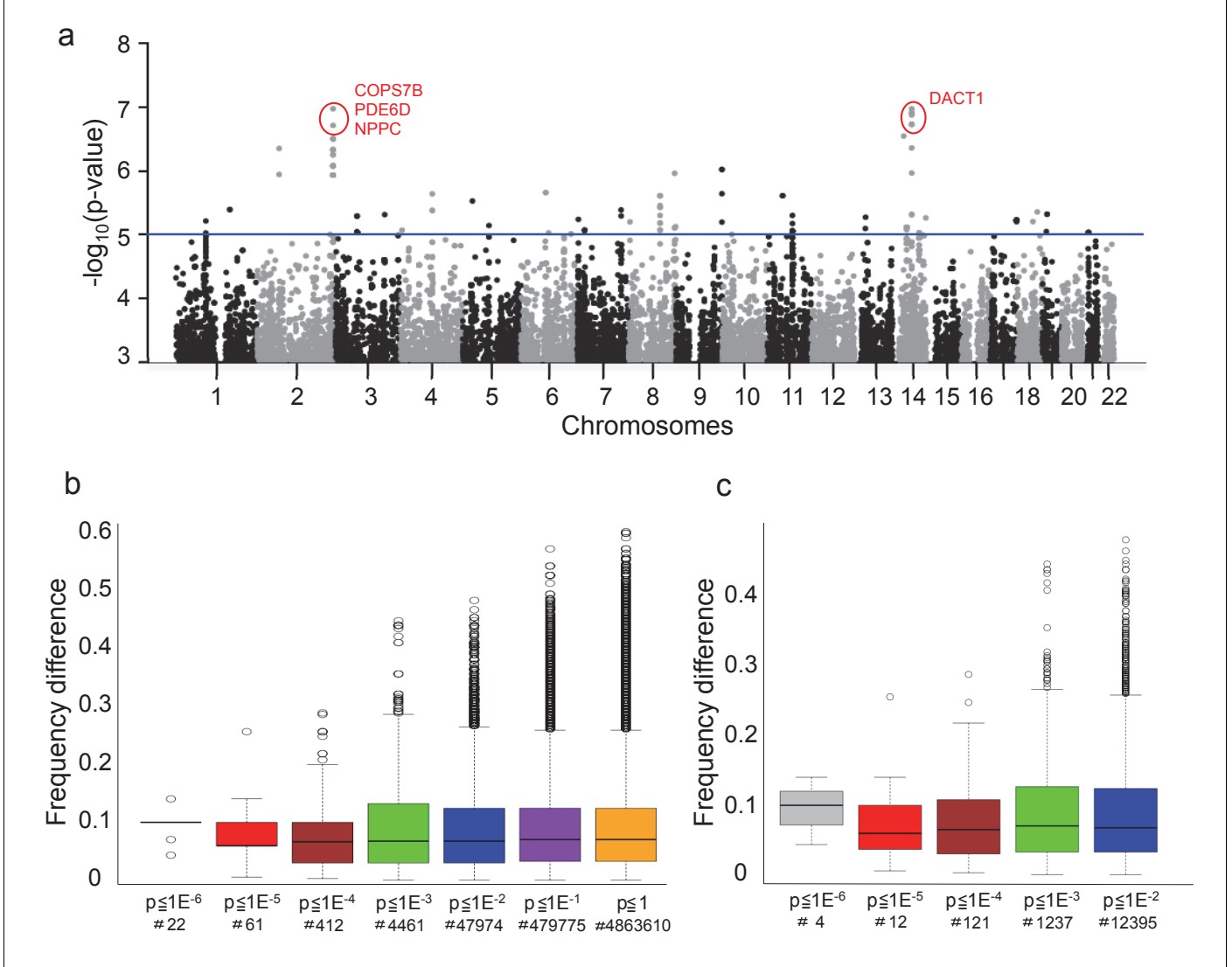

**Figure 1.** eQTLs influencing circadian function. (a) Manhattan plot of identified polymorphisms for all measured traits. Blue line, threshold for suggestive candidates, $p<10^{-5}$, approx. FDR $\leq$ 0.1. Genes likely associated with the most significant alleles are shown in red. (b) For all SNPs at the indicated p-value (x-axis), Tukey Boxplot of the distribution of difference in allele frequency between self-declared 'larks' and 'owls' (y-axis, absolute value of (frequency in larks – frequency in owls); x-axis, p-value threshold used for the test; number of alleles at each stringency indicated underneath). (c) Equivalent chart for Tag-SNPs only, that is one SNP per hapblock. See also *Figure 1—figure supplement 1* for distributions and Manhattan plots for individual traits; see *Figure 1—figure supplement 2* for quantile-quantile plots associated with each Manhattan plot; see *Figure 1—figure supplement 3* for density plots of allele frequency distributions in extreme chronotypes. *Figure 1—source data 1* lists all alleles identified with $p<10^{-5}$ for each trait, as well as the closest gene.

DOI: https://doi.org/10.7554/eLife.24994.002

The following source data and figure supplements are available for figure 1:

**Source data 1.** SNPs associated with (a) *PER1* and (b) *PER2* expression, (c) Amplitude, (d) Phase, and (e) Period.
DOI: https://doi.org/10.7554/eLife.24994.006

**Figure supplement 1.** Umbilical cord fibroblasts from different individuals show large variations in circadian parameters.
DOI: https://doi.org/10.7554/eLife.24994.003

**Figure supplement 2.** Quantile-quantile plots for GWAS of each clock parameter.
DOI: https://doi.org/10.7554/eLife.24994.004

**Figure supplement 3.** 'Larks' and 'owls' show skewed distribution of GWAS-positive alleles.
DOI: https://doi.org/10.7554/eLife.24994.005

While multiple suggestive alleles were identified, it should be noted that none achieved 'genome wide' significance (often reckoned at $p < 5 \times 10^{-8}$), as might be expected from a cell-based study of moderate sample size and 2.5 million SNPs tested. Therefore, we undertook further experiments to demonstrate the relevance of our results.

## Putative significant alleles are enriched in extreme chronotypes

To explore the significance of our results for human behavior and to probe the confidence levels required to elicit meaningful data, we examined the enrichment of SNP alleles in individuals of extreme chronotype after genotyping a previously-characterized cohort of 35 'larks' and 'owls" (*Brown et al., 2008*). We reasoned that for SNPs identified at increasing stringency of p-value, those affecting behavior would show an allelic frequency that varied between larks and owls. Consistent with this hypothesis, allele enrichment analysis shows that differential allelic frequency among SNPs identified at relaxed confidence levels up to $p > 10^{-4}$ is very low, and then increases at $p < 10^{-5}$ or $p < 10^{-6}$, plotted either for all significant SNPs (*Figure 1b*), or as single tag-SNPs per hapblock (*Figure 1c*). The same data is represented as a density distribution in *Figure 1—figure supplement 3*. Here, it can be seen that whereas most alleles have a differential frequency of 0 between larks and owls, this rises progressively to 0.1 for the 22 alleles identified as significant at a threshold of $p < 10^{-6}$.

## Genes associated with positive SNPs affect circadian clock function

As a next step, for each SNP with FDR < 0.1, (using multiple-comparison correction for the 2.5 million SNPs tested) we identified putative genes whose transcription start site lay within 100 kb of the indicated SNP (76 genes). We chose this threshold to correspond to 'suggestive' significance (uncorrected $p < 1 \times 10^{-5}$) in genome-wide association: since it is estimated that only a tenth of the SNPs on the high-density chip that we used would be necessary to capture all common variation in individuals of European origin (*International HapMap Consortium, 2005*), a corrected FDR is therefore up to tenfold 'over-corrected'. We therefore preferred to select genes at a relatively relaxed FDR q-value, and subsequently apply further criteria.

For genes more than 50 kB away from a significant SNP, the further criterion of significant association of the SNP region with the gene of interest via chromatin conformation capture (3C) was applied, using the HiView fibroblast dataset (*Xu et al., 2016*). In total, a list of 59 genes putatively affecting human chronotype was obtained in this way. For example, the two most significant SNPs uncovered in our screen, rs920400 ($p = 1.0E^{-7}$) and rs10195385 ($p = 1.9E^{-7}$) affecting the expression of *PER2*, lay in the conserved enhancer region of the *COPS7B* gene, encoding a subunit of the human COP9 signalosome (*Fang et al., 2008*). These SNPs show a strong 3C signal with the *COPS7B* gene region (*Figure 2a*), supporting their annotation as affecting *COPS7B* expression. (Please see later results for more detailed biochemical and genetic analysis of the effects of these two alleles.)

A second way of looking for associations between SNPs and genes relies upon correlating SNPs with changes in transcription (Regulatory Trait Concordance). Since transcriptomic data from these cells were already available (*Gutierrez-Arcelus et al., 2015*), we used it in order to also search for potential cis- and trans-effects via Regulatory Trait Concordance (*Nica et al., 2010*). This procedure uncovered one additional gene (PPM1B) at these significance criteria. (All genes and their corresponding SNPs are listed in *Supplementary file 2*).

Surprisingly, none of the significant SNPs were present in 'canonical' clock genes of the circadian transcription-translation feedback loop. Therefore, to verify the significance of our findings, we conducted RNA interference (RNAi) targeting each of the genes that we identified using the human model circadian cell line U2OS, and looked for changes in circadian period and amplitude in vitro. Of these 60 genes, depletion of 28 genes showed a significant difference in circadian period of at least one standard deviation from the Z-scored mean, and 29 genes showed similar alterations in circadian amplitude. (Data for all genes tested and representative circadian profiles are shown in *Figure 2—figure supplement 1a,b*). To quantify the significance of these results, we used Redundant siRNA Activity (RSA) scores to attribute an overall p-value to siRNAs tested for each gene (*König et al., 2007*). Thus, for each clock trait – period and amplitude – we obtained two distributions: p-value probability of increase, and p-value probability of decrease of these traits for each

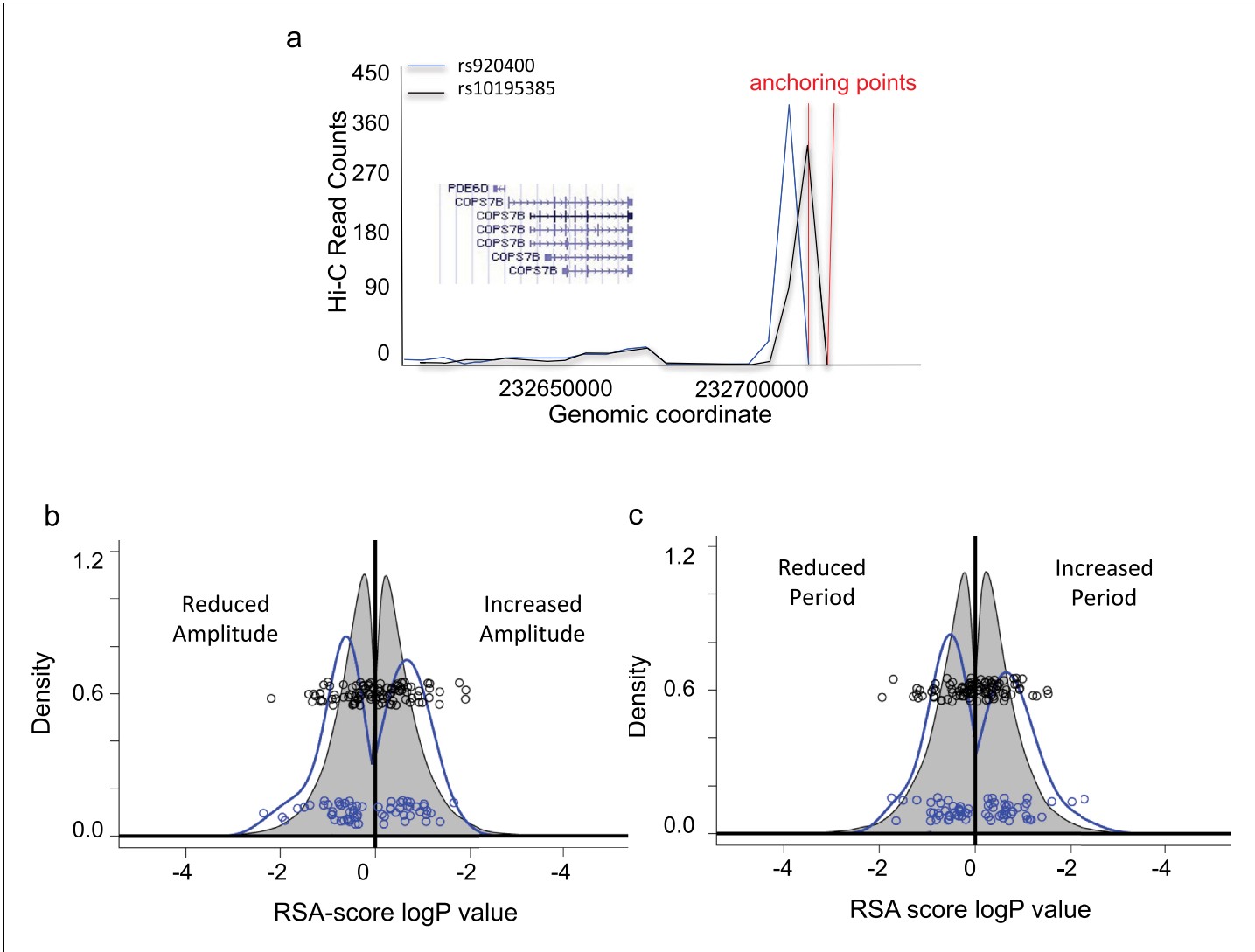

**Figure 2.** Assessment of genes associated with identified eQTLs. (**a**) Genome-wide high-throughput chromosome conformation capture (Hi-C) for probes spanning rs920400 (blue) and rs10195385 (black). Y-axis, read counts. X-axis, genomic location of reads. Gene neighborhood is shown in the inset. (**b**) RSA-score distribution of period lengths of U2OS cell cultures transfected with RNAi hairpins targeting each gene in the genome (grey) or in significant genelist (blue). Y-axis, relative density (area sums to 1); X-axis, log RSA score. Left of 0, period length less than normal; right of 0, period length greater than normal. Blue circles, co-plotted scatterplot of individual values for significant genelist; black circles, corresponding sample set drawn randomly from the whole genome. (**c**) RSA-score density distribution of amplitude. (X and Y axes as in (**b**).) An annotated Significant Genelist is given in *Supplementary file 1*. See also *Figure 2—figure supplement 1* for circadian clock properties of cells in which significant genes are targeted by RNAi, as well as representative raw data and cumulative density plots.

DOI: https://doi.org/10.7554/eLife.24994.007

The following figure supplement is available for figure 2:

**Figure supplement 1.** Clock properties of U2OS cells after RNAi depletion of GWAS-identified transcripts.

DOI: https://doi.org/10.7554/eLife.24994.008

gene identified in our screen. We then compared these to probabilities obtained for the transcriptome as a whole. Overall, the distribution of p-values for period obtained from hairpins targeting genes identified in our screen shifts outward (toward greater significance) and broadens, that is these genes have an increased probability of affecting the circadian clock compared to random genes (*Figure 2b*, Kruskal-Wallis Test p=0.14 reduced and p=0.0016 increased period; the combined cumulative distribution is shown in *Figure 2—figure supplement 1c*, Kolmogorov-Smirnov Test p=0.07). Analogous effects, but stronger, are observed for amplitude (*Figure 2c*, *Figure 2—*

*figure supplement 1c*, p=0.0002 reduced and p=0.034 increased, p=0.0013 combined). Interestingly, a skew to the left is also visible though not significant (*Figure 2—figure supplement 1c*), perhaps reflecting the greater probability of reducing circadian amplitude by depleting identified genes than increasing it.

## Positive SNPs highlight protein catabolism as essential to human circadian variation

To try to understand the broad biochemical pathways underlying the genes that were identified, we first applied gene ontology enrichment analysis (*Wang et al., 2013*) to the list of 60 genes that we obtained in *Supplementary file 2*. The results, depicted as an abbreviated Gene Ontology diagram in *Figure 3a* (full diagram shown in *Figure 3—figure supplement 1*; individual genes and categories listed in *Figure 3—source data 1*), highlight in particular signaling, development, and macromolecular metabolism as significant, with half of all positive genes involved in metabolic processes (*Figure 3b*). To independently confirm these relationships and explore them in more detail, we next used the completely independent approach of a gene set enrichment algorithm (GSEA) adapted for GWAS studies, iGSEA4GWAS (*Zhang et al., 2010*). This algorithm begins by attributing genes to positive SNPs using a distance-based criterion (the same 100 kb that we used previously), looks for pathways with high proportions of significant genes at relaxed stringency criteria (p<$10^{-4}$), and imputes a global significance to this association weighted by SNP p-values of member genes. Applying this method to our SNP data, several pathways of potential significance emerge, including Hedgehog signaling, Cellular Oxidoreductase Activity, Cell Cycle G1/S, and Transcriptional Repressor Activity. A list of these genes and SNPs is presented as *Supplementary file 3*. (A list of categories, all genes in each, and significant genes in each is available in *Figure 3—source data 2*). Five of the top six identified pathways are involved in macromolecular catabolism. The full list of pathway annotations from GSEA analysis is shown in *Figure 3—figure supplement 2a*. We also show an abbreviated list, eliminating redundancies in pathway annotations, in *Figure 3c*.

Since by far the most significant identified category (5 out of the top 6 terms) was protein catabolism, we chose to explore further the significance of alleles in this group. All alleles and their affected genes from this category are shown on a Manhattan plot in *Figure 3d*, and listed in *Figure 3—figure supplement 2b*. The RSA score distribution of these genes showed an increased probability of long- and short-period phenotypes – seen as an outward-shifted and broadened distribution (p=0.009 overall, 2-way Kolmogorov-Smirnov Test; p=0.044 shorter, p=0.0008 longer, Kruskal-Wallis Test; *Figure 3e*, *Figure 3—figure supplement 2d*), and a significant skew toward reduced circadian amplitude (p=0.024 reduced, p=0.47 increased, Kruskal-Wallis Test, *Figure 3f*, more easily seen as a cumulative distribution in *Figure 3—figure supplement 2e*). Thus, the results of our genetic screen support the hypothesis that a major reason for human circadian variation at a cellular level relates to global alterations in protein stability.

## COPS7B *influences PER2 expression and clock function.*

Consistent with this idea, the top two SNPs identified in our screen are in putative regulatory regions of the COPS7B gene (as well as 24 other alleles in the same hapblock with p<$10^{-6}$). This protein is part of the COP9 signalosome, which has been implicated both in regulating ubiquitin-mediated proteolysis (*Pick and Bramasole, 2014*), and in regulating circadian behavior of *Neurospora crassa* and *Drosophila melanogaster* (*He et al., 2005*; *Knowles et al., 2009*). Therefore, understanding how COPS7B affects circadian function could provide important clues to the mechanisms of genetic influence upon human circadian clocks.

To access the effect of *COPS7B* allelic identity upon expression of a clock gene, we independently measured *PER2* expression values as a function of rs920400 genotype, and found that average *PER2* expression in the rs920400 AA genotypes were significantly higher (p=0.00026) than the GG genotype (*Figure 4a*).

As mentioned previously, this SNP falls in a conserved putative enhancer region for its closest gene, *COPS7B*, lying 30 kb away. To demonstrate a potentially functional relationship between COPS7B and PER2, for each allele we plotted the relationship between *COPS7B* expression and *PER2* expression, and demonstrated strong positive correlations within each genotype (AA: r = 0.9327, p=0.0007; AG: r = 0.9455, p=0.0004, GG: r = 0.8666, p=0.0054) (*Figure 4a*, insets).

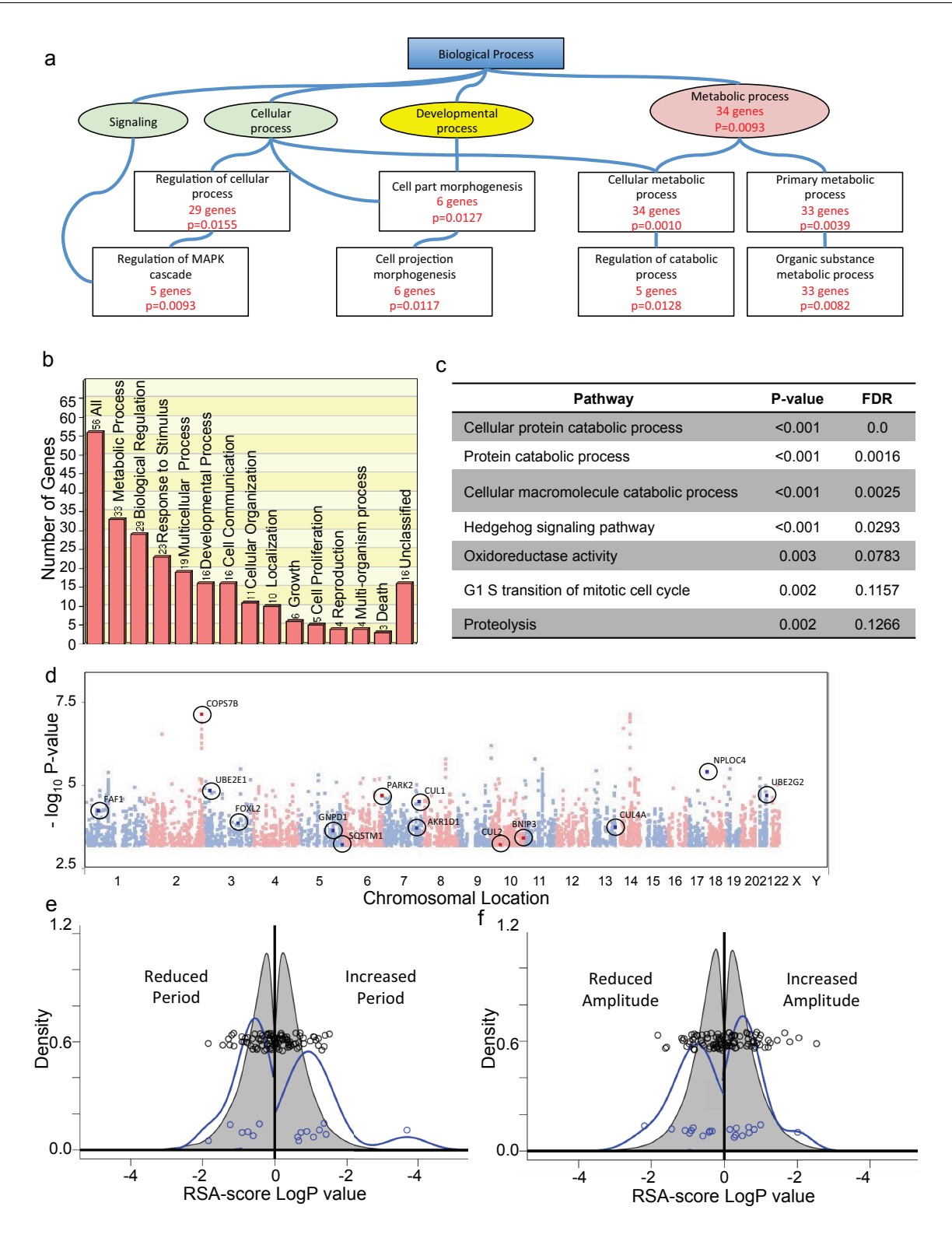

**Figure 3.** Positive SNPs highlight protein catabolism as essential for human circadian variation. (a) Abbreviated Gene Ontology diagram from the Gene SeT Analysis Toolkit (WebGestalt), showing biological process categories containing genes (*Supplementary file 1*) associated with positive SNPs. Boxes with red statistical description represent enriched categories, with the number of associated genes and corresponding multiple comparison-adjusted p-value. (Individual pathways and associated genes are annotated in *Figure 3—source data 1*.) (b) Biological processes categorization of all

*Figure 3 continued on next page*

*Figure 3 continued*

identified genes (GOSlim tool). Y axis, number of genes in the category. (**c**) The most significant nonredundant pathways associated with positive alleles, identified by GeneSet Enrichment Analysis (GSEA4GWAS toolkit; a full chart including closely-related pathways is given in *Figure 3—figure supplement 1*; All genes are listed in *Supplementary file 2*; individual pathways and a clickable list of associated genes and SNPs is given in *Figure 3—source data 2*.) (**d**) Manhattan plot of all alleles p>10$^{-3}$. GSEA-uncovered genes related to protein catabolism are indicated, and associated polymorphisms are circled. (**e**) RSA-score distribution of period lengths of U2OS cell cultures transfected with RNAi hairpins targeting each gene in the genome (grey), or targeting genes in (**d**) (blue). Circles, co-plotted scatterplot of individual values. X and Y axes as in *Figure 2b,c*. (**f**) RSA score distribution for circadian amplitude observed in U2OS cell cultures transfected with RNAi hairpins targeting each gene in the genome (grey), or targeting genes in (**d**) (blue). Circles, co-plotted scatterplot of individual values. X and Y axes as in *Figure 2b,c*. See also *Figure 3—figure supplement 1* for a full list of GSEA-identified pathways, as well as a list of RNAi-targeted genes associated with protein catabolism, and the circadian clock properties of the cells in which this targeting occurred.

DOI: https://doi.org/10.7554/eLife.24994.009

The following source data and figure supplements are available for figure 3:

**Source data 1.** Multiple-comparison-adjusted p-values and significant associated genes for each pathway identified in *Figure 3a* (and in more detail in *Figure 3—figure supplement 1*).
DOI: https://doi.org/10.7554/eLife.24994.012

**Source data 2.** Multiple-comparison-adjusted p-values, false discovery rates, and significant associated genes for each pathway identified in *Figure 3—figure supplement 2a*.
DOI: https://doi.org/10.7554/eLife.24994.013

**Figure supplement 1.** Full Gene SeT Analysis Toolkit (WebGestalt) biological process category diagram containing genes associated with positive SNPs.
DOI: https://doi.org/10.7554/eLife.24994.010

**Figure supplement 2.** Clock properties of U2OS cells after RNAi depletion of GSEA-identified transcripts.
DOI: https://doi.org/10.7554/eLife.24994.011

It should be noted that here and in the screen itself, a single timepoint was used to investigate *PER2* expression, corresponding to its expected first peak after cell line synchronization. Thus for any single subject, absolute changes in level could also reflect changes in timing. Therefore, we next explored the origins of this effect more fully using exogenous assays. We transfected the circadian model cell line U2OS:*Bmal1-luc* (*Maier et al., 2009*) with three different short interfering RNAs targeting *COPS7B*, reducing its expression by 68%. These siRNAs correspondingly reduced *PER2* expression (*Figure 4b*), and also decreased circadian amplitude more than 10-fold, p=2.93×10$^{-7}$ and lengthened circadian period by 1.15 hr, p=3.27×10$^{-5}$ (*Figure 4c,d*), even as cells continued to divide viably. Showing that the modulation of circadian clock function is achieved by the COP9 signalosome rather than by COPS7B independently, a an even more severe phenotype was achieved by targeting *COPS4*, an adjacent COP9 signalosome subunit: circadian amplitude was reduced more than 10-fold in the first day, p=3.59E$^{-7}$ and period initially lengthened by 8.8 hr (p=4.02E$^{-5}$), followed by arrhythmicity, (*Figure 4—figure supplement 1a,b*). Because this phenotype is the opposite of the shorter period observed by eliminating either PER protein directly (*Zheng et al., 2001*), it also led us to suspect that the effects upon *PER2* transcription were indirect.

## COPS7B interacts physically with clock proteins and stabilizes BMAL1

To demonstrate physical interactions of the COP9 signalosome with the circadian clock machinery, we transfected HEK 293 T cells – which do not possess a functional circadian clock, thereby eliminating the variable of time – with constructs expressing FLAG epitope-tagged clock proteins BMAL1, PER2, and CRY1. Subsequently, each lysate was immunoprecipitated with either anti-FLAG or anti-COPS7B antibody, and then Western-blotted and probed with the opposite. Interactions between COPS7B and all three clock proteins were observed bidirectionally (*Figure 5a,b*), while no corresponding clock proteins were immunoprecipitated by FLAG antibody in the absence of transfection (*Figure 5—figure supplement 1*).

The COP9 signalosome has been implicated previously in antagonizing ubiquitin-mediated degradation of proteins (*Korczeniewska and Barnes, 2013*; *Fernandez-Sanchez et al., 2010*). Therefore, we next measured the half-life of transfected FLAG-tagged circadian clock proteins by metabolic pulse-chase protein labeling in the presence and absence of siRNAs targeting COPS7B. The time-courses of degradation of CRY1 and PER2 remained unchanged (*Figure 5—figure supplement 2*), but the half-life of BMAL1 was reduced to 4.4 hr when compared to control whose half-life was 7.3

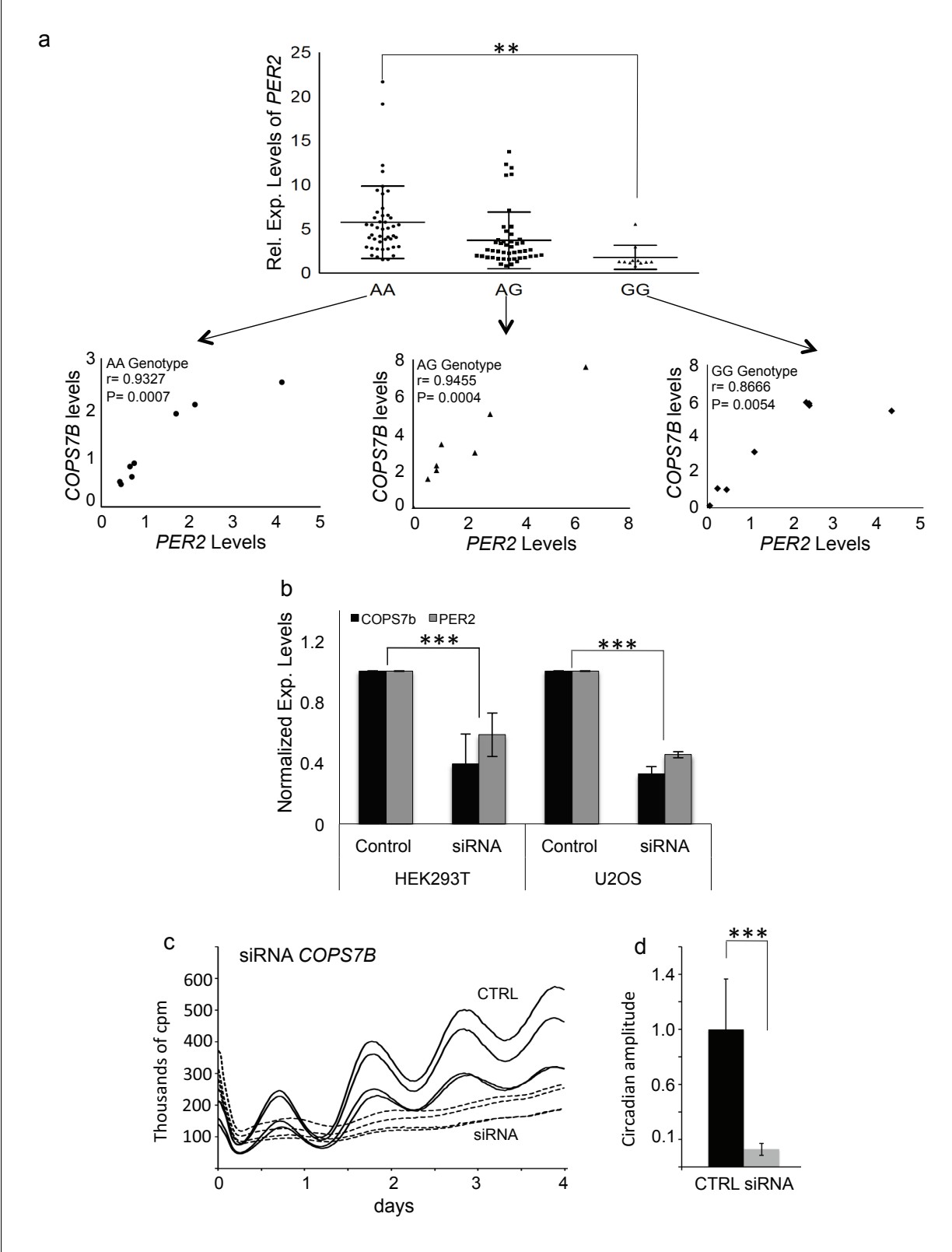

**Figure 4.** COPS7B influences *PER2* expression and cellular clock function. (a) Expression levels of *PER2* stratified by rs920400 genotype. (AA vs. GG genotype, T-test p=0.00026). **Insets** - Correlation between *PER2* and *COPS7B* gene expression within each genotype. (b) Reduction of *PER2* expression (grey) and *COPS7B* expression (black) in HEK293T and U2OS cells transfected with three short interfering RNAs targeting *COPS7B* (c) Raw bioluminescence profiles of U2OS cells cotransfected with a *Bmal-luciferase* circadian reporter and siRNAs targeting COPS7B (dotted lines) or
*Figure 4 continued on next page*

*Figure 4 continued*

scrambled siRNA (solid lines). (**d**) Reduction in circadian amplitude by short interfering RNAs targeting *COPS7B* (p=0.0021, student T-test). These siRNAs also lengthen the circadian period by 1.15 hr (p=3.27094E$^{-05}$). See also *Figure 4—figure supplement 1* for similar RNAi assays performed upon the *COPS4* gene.

DOI: https://doi.org/10.7554/eLife.24994.014

The following figure supplement is available for figure 4:

**Figure supplement 1.** Effects of COPS4 depletion upon circadian period and amplitude.

DOI: https://doi.org/10.7554/eLife.24994.015

hr. (*Figure 5c*). Therefore, we hypothesized that the *bona-fide* interaction between COP9 and the circadian clockwork was with BMAL1, even if biochemical interactions with other clock proteins could be found in transfection-based assays..

## COPS7B and the COP9 signalosome are rhythmically imported into the nucleus

Having demonstrated an effect of COP9 signalosome impairment upon BMAL1 protein stability in HEK 293 T cells, we next examined the same phenomenon in U2OS cells, which possess robust circadian rhythmicity. In these cells, interaction was observed with BMAL1 only (*Figure 5d*). A subsequent circadian timecourse using synchronized cells harvested at different times of day illustrated one possible cause for the selectivity of this interaction: by Western blot, COPS7B shows circadian nuclear abundance in phase with BMAL1 but antiphase to PER and CRY proteins (*Figure 6a*). It should be noted that this result does not completely explain why interactions with other clock proteins would not be observed when they are exogenously transfected, and suggests therefore that time-specific cofactors might be required for the interaction.

To demonstrate the relevance of our findings in vivo, we turned to a mouse liver model. Examining a mouse liver transcriptomic dataset (*Atger et al., 2015*), the RNAs encoding most COP9 signalosome subunits showed arrhythmic transcription, but *Cops7b* transcripts were markedly circadian (*Figure 6b*), and also expressed at the lowest level (*Figure 6—figure supplement 1a*). Since our COPS7B antibodies were specific to human, we used isotope-labeled mass spectrometry to examine nuclear accumulation of COP9 signalosome components. All COP9 subunits are imported rhythmically into liver nuclei in phase with the peak of BMAL1 protein accumulation, and slightly in advance of it (*Figure 6c*), implying a circadian role for COP9 at this time of day. Thus, both rodent and cellular studies support the idea that changes in COP9 expression could alter circadian function.

## Discussion

A salient feature of human circadian behavior is its difference across individuals.

Previous studies have focused upon the importance of 'canonical' clock genes – core members of the circadian transcription-translation feedback loop – in these variations. Specific clock gene mutations been identified both in families (*Toh et al., 2001*; *Xu et al., 2005*) and in the broader population (*Allebrandt and Roenneberg, 2008*; *von Schantz, 2008*) that could affect human behavior. Moreover, three recently-published large questionnaire-based GWAS studies of circadian behavior have also highlighted polymorphisms in the region of known clock genes as a sizeable portion of the signals they observe (*Hu et al., 2016*; *Lane et al., 2016*; *Jones, 2015*). Taking a cellular rather than a behavior-based GWAS approach to circadian variation, we find evidence of a wider biochemical truth: a major source for differences in clock function at a cellular level is in fact based upon the broad mechanism of protein catabolism. Surprisingly, there is no overlap between top candidate alleles in their studies and ours. Moreover, among these three studies, so far only meta-analysis of grouped data has been performed, and no evidence was seen for other previously implicated alleles. Thus, the possible space for novel discovery remains large. Moreover, it is unclear to what extent any given QTL affecting cellular clock properties such as those that we investigate would propagate directly to effects upon human behavior.

As a feedback loop, the circadian clock is critically dependent upon protein stability to time its oscillations. Therefore, our conclusions make mechanistic sense. Indeed, both genetic linkage studies cited above (*Toh et al., 2001*; *Xu et al., 2005*) have highlighted the importance of post-translational

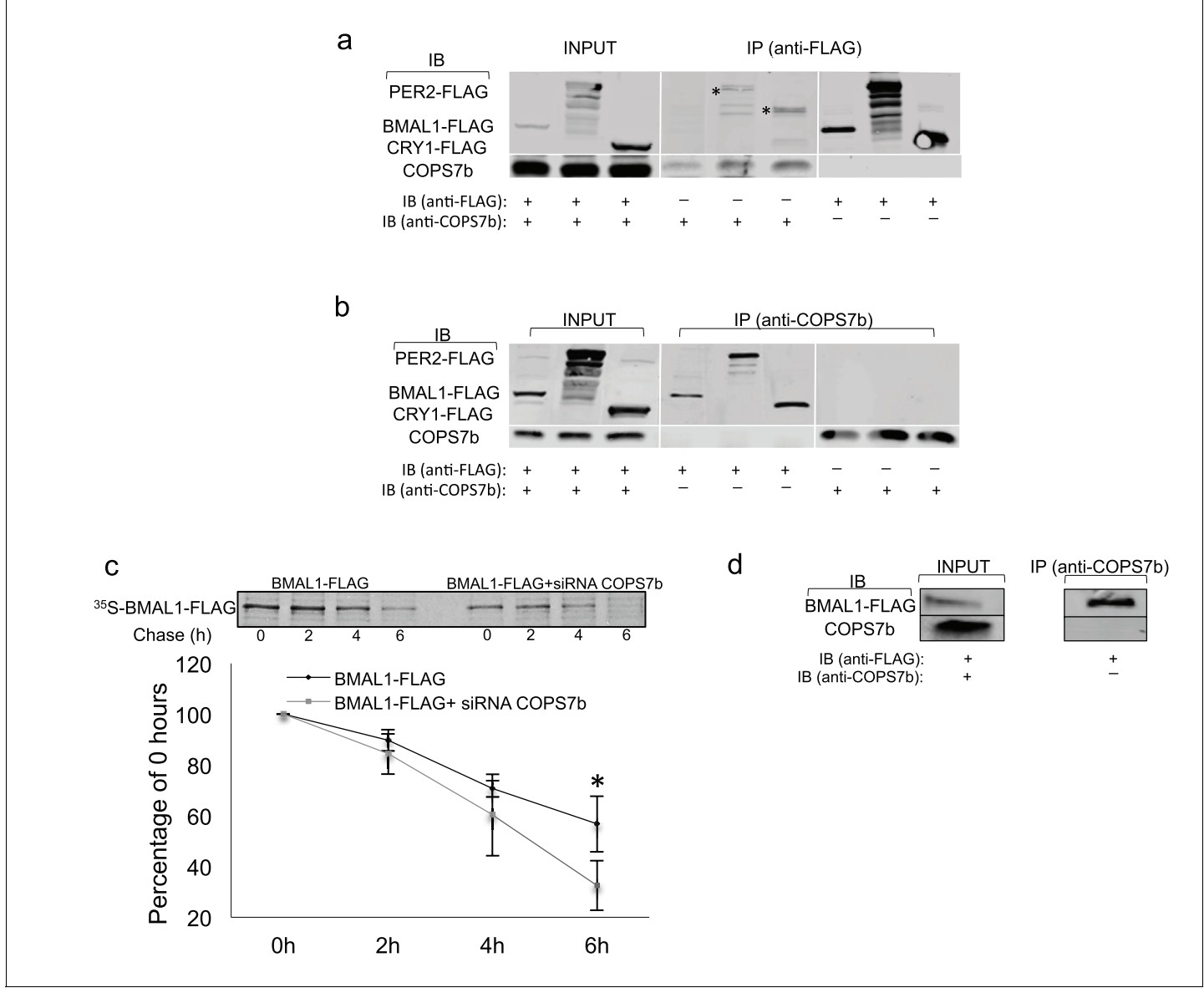

**Figure 5.** COPS7B stabilizes the essential clock protein BMAL1. (**a–b**) Coimmunoprecipitation assay of HEK293T cells transfected with plasmids expressing FLAG epitope-tagged clock proteins (BMAL1-flag, PER2-flag and CRY1-flag). *Unspecific background bands. (**a**) immunoprecipitated with antibodies against FLAG or (**b**) against endogenous COPS7B. (**c**) Pulse-chase analysis of BMAL1-flag protein stability in absence (left; black diamonds in graph below) or presence (right; grey squares in graph below) of siRNAs targeting COPS7B in HEK293T cells. All cells were incubated with $^{35}$S-labelled methionine-cysteine for 1 hr, and chased with excess unlabeled cysteine for 0 hr, 2 hr, 4 hr, or 6 hr before immunoprecipitation. Upper panel, representative radioblot; lower panel, quantification from 3 experiments ± s.d., expressed as percentage of labeled immunoprecipitated protein (relative to 0 hr) at indicated time. *p=0.04999, Student T-test. (**d**) Immunoprecipitation experiments between BMAL1 and COPS7B performed in U2OS cells. See also *Figure 5—figure supplement 1* for negative control experiments upon untransfected cells; see *Figure 5—figure supplement 2* for pulse-chase analysis of PER2-flag and CRY1-flag protein stability in the presence or absence of siRNAs targeting *COPS7B*.

DOI: https://doi.org/10.7554/eLife.24994.016

The following figure supplements are available for figure 5:

**Figure supplement 1.** Negative control immunoprecipitation experiments identical to those described in *Figure 5a,b*, but using untransfected HEK293T cells as a substrate.

DOI: https://doi.org/10.7554/eLife.24994.017

**Figure supplement 2.** Pulse-chase analysis of PER2 (**a**) and CRY1 (**b**) protein half-lives in absence (left-control lanes; black diamonds in graph below) or presence (right lanes; grey squares in graph below) of siRNAs targeting *COPS7B*.

DOI: https://doi.org/10.7554/eLife.24994.018

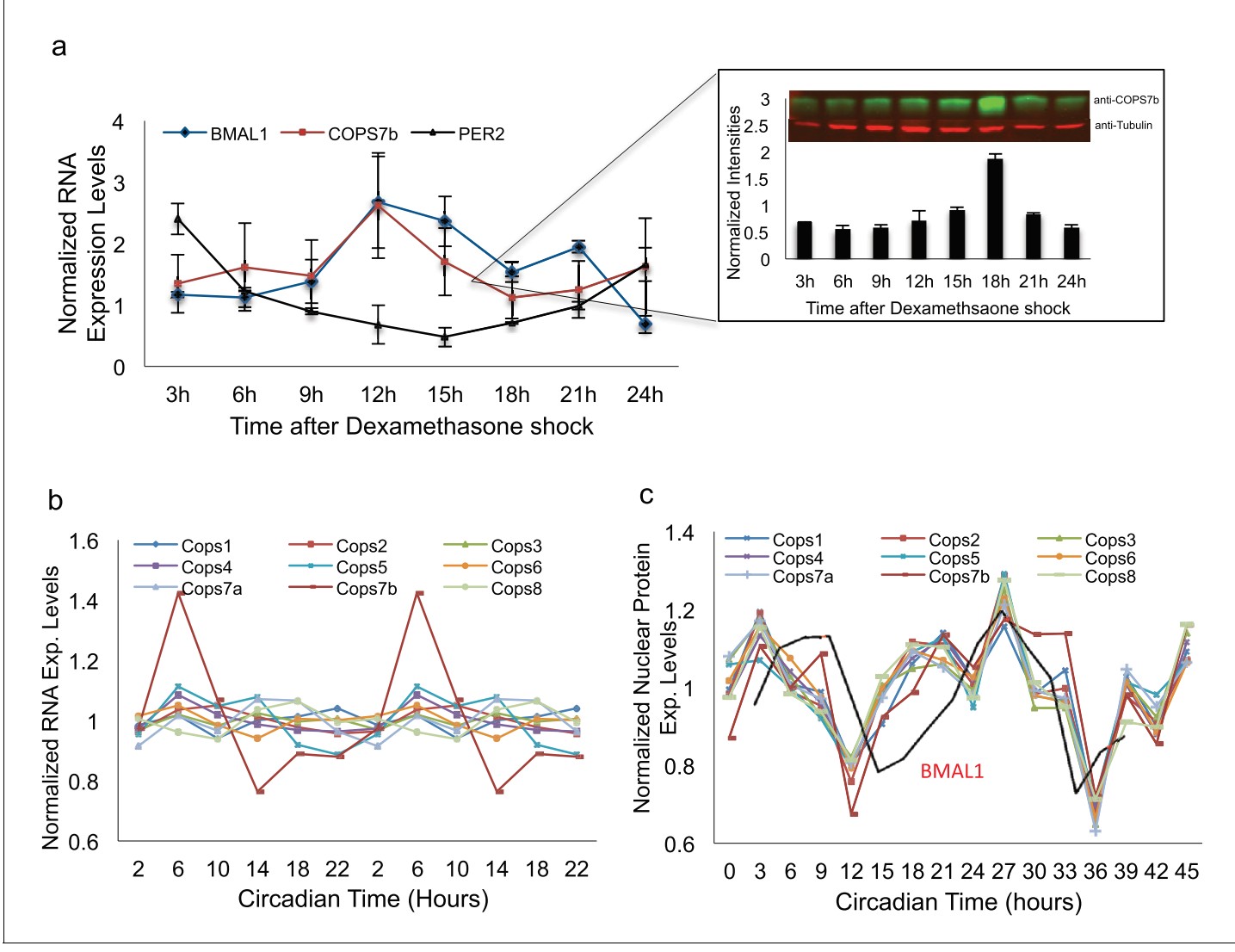

**Figure 6.** The signalosome is rhythmically imported into the nucleus. (**a**) *BMAL1*, *PER2* and *COPS7B* RNA levels in U2OS cells at different times of day, as determined via qPCR. **Inset** - Western blot analysis of nuclear COPS7B protein under identical conditions. Top panel, representative raw blot; bottom panel, quantification. (**b**) RNAseq transcript levels of genes encoding COP9 signalosome subunits in mouse liver at different times of day. (**c**) Protein levels of COP9 signalosome subunits measured by MS in mouse liver nuclei at different times of day. Red line, protein levels of circadian clock protein BMAL1. See also *Figure 6—figure supplement 1* for ms-based relative abundance analysis of different COP9 signalosome subunits, as well as protein levels of proteasome subunits measured by MS in mouse liver nuclei at different times of day.

DOI: https://doi.org/10.7554/eLife.24994.019

The following figure supplement is available for figure 6:

**Figure supplement 1.** Relative expression of signalosome and proteasome subunits.

DOI: https://doi.org/10.7554/eLife.24994.020

modification of clock proteins to regulate ubiquitin-proteasome pathway-dependent protein stability. RNAi screens (*Maier et al., 2009*; *Sathyanarayanan et al., 2008*; *Zhang et al., 2009*) and mouse mutant analyses (*Godinho et al., 2007*; *Siepka et al., 2007*) have also emphasized the global importance of proteasome-mediated degradation to circadian function. Here, by using human genetics as a tool for mechanistic discovery, we have demonstrated a novel role for the COP9 signalosome in the stability of the essential circadian clock protein BMAL1, a transcriptional activator of the *PER2* gene (*Gekakis et al., 1998*) and a critical player in the circadian process. In fact, the entire signalosome is imported into the nucleus both in vitro and in vivo in time with the maximal abundance of BMAL1, enhancing its stability. Thus, the primary effect is a competition between signalosome and

proteasome to stabilize or destabilize BMAL1, the former in time-specific fashion. Although we could detect interactions with multiple transfected clock proteins in nonrhythmic HEK cells, in rhythmic U2OS cells only BMAL1 interaction was detectable, suggesting further that other time-of-day-specific factors might be required for this interaction.

For the 'negative limb' of the circadian clock, such a phenomenon has already proven essential to the regulation of circadian period. Competition between two different E3 ubiquitin ligases, FBXL3 and FBXL21, respectively destabilize and stabilize CRY proteins (*Hirano et al., 2013*; *Yoo et al., 2013*). Analogous effects can be observed for PER2 protein, this time via competition between different phosphorylation events (*Vanselow et al., 2006*; *Xu et al., 2007*). Competition for phosphorylation sites and ubiquitin ligases is also a key determinant of the speed of the Drosophila circadian oscillator (*Chiu et al., 2011*; *Chiu et al., 2008*). With these genetic studies, we have unearthed evidence of a conceptually similar but mechanistically different regulation of the 'positive limb' via BMAL1 stability.

It is worth noting that although knockdown of COP9 subunits altered circadian period, phase, and *PER2* expression, the *COPS7B*-associated alleles rs920400 and rs10195385 that we genetically identified were associated significantly only with *PER2* expression in vitro: rs920400/PER2: $p=1.0677E^{-7}$, rs10195385/PER2: $p=1.9463E^{-7}$. For the same SNPs, association with other clock properties was not significant – *rs920400/PER1: p=0.202572306, rs10195385/PER1: p=0.198630802; rs920400/AMPLITUDE: p=0.791990033, rs10195385/AMPLITUDE: p=0.944165947; rs920400/PHASE: p=0.489070695, rs10195385/PHASE: p=0.522821404; rs920400/PERIOD: p=0.602715502, rs10195385/PERIOD: p=0.700789177*. More generally, although there was detectable and significant genotypic correlation between period and phase, and *PER1* and *PER2* expression in our screen (*Supplementary file 1*, *Figure 1—figure supplement 3*), genetic correlation among these different phenotypes was nevertheless weaker than we expected. We attribute this weakness to the idea that small changes in cellular clock properties can be robustly tolerated without affecting other clock parameters. Such an idea has also been supported by cellular dosage studies individual clock proteins (*Baggs et al., 2009*).

The COP9 signalosome has been shown previously to inactivate many cullin-RING-ubiquitin E3 ligases, probably by removing ubiquitin-like NEDD8 from cullins (so-called 'de-neddylation') (*Dubiel et al., 2015*; *Cope et al., 2002*). It has also been demonstrated to interact with casein kinase 2 (*Uhle et al., 2003*), which itself has a circadian role in mammals (*Maier et al., 2009*). Since the proteasome itself is constitutively present in the nucleus (*Figure 6—figure supplement 1b*) but the signalosome is circadian, it could thereby provide a more general mechanism for circadian regulation of nuclear protein stability. For example, in other eukaryotes, it has been suggested that the COP9 signalosome plays a role in the ubiquitin-mediated degradation of the clock protein TIMELESS in *Drosophila melanogaster* (*Knowles et al., 2009*), and Frequency in *Neurospora crassa* (*He et al., 2005*). Notably, the role of COP9 in *Drosophila* is also stabilizing, but opposite in effect: by stabilizing the ubiquitin ligase itself, COP9 thereby promotes degradation of TIMELESS.

Mechanisms related to protein stability are the most prominent identified here – not only the signalosome, but also kinases, phosphatases, ubiquitin ligases, and regulatory proteins. However, it was not the only one. For example, the Hedgehog signaling pathway is represented at high significance, not only through casein kinase 1 homologs – a protein family known to have circadian function (*Cheong and Virshup, 2011*) – but also through WNT secreted signaling molecules. These were recently shown to play a role in cell cycle-circadian clock synchronization in intestinal organoids (*Matsu-Ura et al., 2016*). Cell cycle proteins are also represented – logically, since the cell division and circadian clock cycles are robustly coupled (*Bieler et al., 2014*), but nevertheless mechanistically not well understood. Likewise, cellular redox has come to recent attention in the context of circadian clocks (*Putker and O'Neill, 2016*), but specific mechanisms remain to be defined. Overall, human genetic studies have considerable power to discover novel physiologically relevant circadian mechanisms, and cell-based approaches make this discovery process even easier. In the case of circadian biology, such investigations could both provide new targets directly relevant to human health, and help to explain clock connections to disease.

# Materials and methods

### Ethical approval

All human samples used in these studies were obtained after approval of all protocols and procedures by the relevant responsible authorities (of the University Hospital Geneva, CH; and Charité Universitätsmedezin, Berlin, DE), and prior written informed consent was obtained from all subjects.

### Tissue isolation, cell culture and genotyping

Umbilical cord fibroblasts were collected from 159 newborns of Western European origin as described previously (*Dimas et al., 2009*). Briefly, under sterile conditions, the cord tissue was finely cut in 1 ml DMEM containing 10% FCS, 1% antibiotics (Amimed, Basel, Switzerland), transferred to a T25 flask and cultured upside-down for 12 hr to allow cells to attach to the surface of the flask. Flasks were then turned and kept for about 1 week until fibroblast clusters appeared. Subsequently, fibroblasts were expanded with standard procedures. Similarly, fibroblasts from Western European adults of extreme chronotype (17 'owls' and 11 'larks') were cultured as described previously (*Brown et al., 2008*).

All fibroblasts were genotyped with the Illumina Omni2.5–8 and the Omni2.5Exome-8 chips. Population stratification was done by principal component analysis with phase 1 1000 genomes variants. The phasing was performed with the SHAPEIT v.2.r790 software (*Delaneau et al., 2011*) and the imputation with the IMPUTE2 v2.3.1 software using the phase 1 1000 genomes genotypes as a reference panel. Variants were filtered out according to these criteria: MAF <5%, Hardy-Weinberg probability <1e-6, call rate <97.5% and imputation score <0.4. A total of 5210911 variants were left after these filtering steps.

### Fibroblast culture, viral transduction, synchronization, and measurement of circadian rhythms parameters

Cultivation of fibroblast cells, virus preparation, transduction, and selection of stable transformants were performed as previously described (*Brown et al., 2005*). Before the measurement, circadian rhythms in identically grown plates were synchronized with the synthetic glucocorticoid dexamethasone (*Balsalobre et al., 2000*). After washing twice with 1x PBS, cells were incubated in DMEM medium without phenol red, supplemented with 10% FBS, 0.1% gentamycin and 0.1 mM luciferin. Light emission was measured in a lumicycle photomultiplier device at 1 min intervals at 37°C, 5%$CO_2$ for 5 days.

### RNA preparation and gene expression quantification

Total RNA was extracted by using High Pure RNA Isolation Kit (Roche) following the instructions of the manufacturer. 500 ng of isolated RNA was transcribed with SuperScript II (Invitrogen, Carlsbad, CA) using random hexamer primers according to the manufacturer's instruction. For quantitative PCR, 20 ng of cDNA was used and single transcript levels of genes were detected either by TaqMan probes used with the TaqMan PCR mix protocol (Roche, Basel, Switzerland) or with HOT FIREPol EvaGreen qPCR Mix Plus (Solice Biodyne, Tartu, Estonia) and an AB7900 thermocycler (Applied Biosystems, Foster City, CA). Primers used for detection of transcripts were as follows: *BMAL1* forward: 5'-gaagacaacgaaccagacaatgag-3', *BMAL1* reverse: 5'- acatgagaatgcagtcgtccaa-3', *BMAL1* TaqMan probe:5'-FAM-tgtaacct-cagctgcctcgtcgca-TAMRA-3'; *PER1* forward: 5'-cgcctaacccctatgtga-3', *PER1* reverse: 5'-cgcgtagt-gaaaatcctcttgtc-3', *PER1* TaqMan probe: 5'-FAM-cgcatccattcgggttacgaagctc-TAMRA-3'; *PER2* forward: 5'-gggcagcctttcgactattct-3', *PER2* reverse: 5' gctggtgtccaacgtgatgtact-3', *PER2* TaqMan probe:5'-FAM-cattcggtttcgcgcccggg-TAMRA-3'; *GAPDH* forward: 5'-cacatggcctccaaggagtaa-3', *GAPDH* reverse:5'-tgagggtctctctcttcctcttgt-3', *GAPDH* TaqMan probe:5'-FAM-tggaccaccagccccag-caaga-TAMRA-3'; *BLASTICIDIN* forward: 5'- gcgacggccgcatct-3', *BLASTICIDIN* reverse: 5'-acaaggtcccccagtaaaatg-3',

*BLASTICIDIN* TaqMan Probe: 5'-FAM-cactggtgtcaatgtat-TAMRA-3' (FAM is 6-carboxyfluorescein, and TAMRA is 6-carboxytetramethylrhodamine);

*COPS7B* forward: 5'-ggctgggattagggtggttc-3', *COPS7B* reverse:5'-atgtgggtacaggccttcctc-3'.

## Data analysis of bioluminescence measurements and transcript levels

The circadian period length, phase, and amplitude were analyzed via the Waveclock R package (*Price et al., 2008*). To normalize circadian amplitude, the relative expression levels of *BLASTICIDIN* determined by qPCR for each subject were used. Data sets for the period length, phase, and amplitude were measured and plotted as average ±standard error from four replicates in three independent measurements for each subject. Peak expression levels of *PER1* and *PER2* likewise the levels of *COPS7B* and *BLASTICIDIN* transcripts were measured in four replicates for each tested subject.

## Phenotype normalization

Prior to GWAS analysis, all data was corrected to eliminate systemic differences related to technical variability from cell passage and date of experiment. First, we averaged data coming from technical replicate of the same biological replicate. Second, we noted that some of our phenotypes (circadian period length, phase and amplitude) varied slightly but globally with the passage number of cell lines as well as with the date of the experiment. In order to eliminate effect of these confounding parameters we residualized our phenotypes using backward multiple regression model (lme function in nlme R package): starting from all variables included into the model we step by step eliminated less significant ones until all variables were nominally significant (p<0.05). Passage number we treated as ordinary numerical variables while date of experiments – as factors (dummy variables). Because all biological replications are non-independent, that is they belong to the same cell line, we used the cell line ID as a grouping variable in the lme function. Third, averaging residualized data coming from each biological replication we obtained the final phenotype value for each cell line. These values were used for all downstream analyses.

## Genome-wide association study and statistical analysis

Independent GWAS were performed on the following normalized phenotypes: *PER1* (140 samples), *PER2* (141 samples), amplitude (142 samples), phase (147 samples) and period (147 samples). The GWAS were performed by using Spearman correlation between the genotypes and the phenotypes (continuous variable). All nominal p-values were reported on the Manhattan plots.

## Regulatory Trait Concordance (RTC)

In order to link our GWAS hits with potential causal variants affecting the expression of genes, we used the RTC method described by Nica et al (*Nica et al., 2010*). GWAS variants with p-values smaller than $1E^{-4}$ were queried by the RTC method on a list of expressed quantitative trait loci (eQTL) found on GENCORD fibroblast samples (*Gutierrez-Arcelus et al., 2015*). The GWAS variants rs1878511 (p-value: $1E^{-4.089}$) and rs34041043 (p-value: $1E^{-4.089}$) had a RTC score bigger than 0.9 indicating a colocalization of the GWAS signals with the eQTL gene: *PPM1B* (SNP: rs4953137).

## Allele enrichment analysis

This method was developed to show that alleles identified as affecting circadian clock properties in vitro would also affect human behavior in vivo. For each genotyped SNP, an allele frequency difference was calculated as the absolute value of (allele frequency in larks – allele frequency in owls). Larks and owls here indicate a group of individuals of extreme early and late chronotype identified in a previous study from our laboratory (*Brown et al., 2008*). The average frequency difference was then calculated for each group of alleles identified as significant in our genetic screen for a given p-value, and graphed either as a bar-and-whiskers plot (*Figure 1b,c*) or as a density plot (*Figure 1— figure supplement 3*).

## RNA Interference assays

Lentiviral vectors expressing RNAi hairpins were purchased from Open Biosystems. Viral production, infection of U-2 OS cells, and subsequent measurement of circadian oscillations was performed as described in (*Maier et al., 2009*), n = 1/construct. Data analyses were also performed as described therein. Z-scored values for period and amplitude were calculated from individual RNAi constructs, and multiple RNAi constructs for a given gene were then concatenated using Redundant SiRNA Activity (RSA) scores calculated as described (*König et al., 2007*). Density distributions of RSA scores were calculated for a given genelist using all hairpins available in the Open Biosystems library,

and compared to that for the entire transcriptome present therein using density, Kolmogorov-Smirnov, and Kruskal-Wallis tests from the R stats package. Density distributions were plotted either as mirror-image half-graphs showing probabilities of increase and decrease on the same axes, or as cumulative distributions of the combined set of both probabilities. (Practically, this was achieved by inverting the sign of the P value distributions reflecting increased period or amplitude.) For display purposes, the entire set of genelist data was shown as open circles, and compared to a random sample of 50 genes randomly chosen from the increased value distribution and 50 from the decreased value distribution. Note that RSA scores for hairpins showing different results for the same gene (e.g. one with increased and one with decreased period) will display a probability in both distributions, so the number of points observed can be greater than the total number of genes.

## Bioinformatics

Chromatin conformation capture data was obtained from the HiView dataset and web-based search engine (*Xu et al., 2016*), and visualized on the UCSC genome browser http://genome.ucsc.edu/. Pathway analysis was done using the Webgestalt query tool (*Wang et al., 2013*) with default settings. GSEA4GWAS (*Zhang et al., 2010*) was used with 100 kB windowing and canonical pathways.

### Cell lines

HEK293T and U2OS cells used for these studies were purchased from Sigma-Aldrich, St. Louis, MO. U2OS:*Bmal1-luc* cells were constructed as described (*Maier et al., 2009*). All lines were verified to be mycoplasma-free by PCR-based testing shortly prior to these experiments.

## Transient transfection protocols

To access possible functional relationship between the COPS7B subunit of the COP9 signalosome and the circadian clock, 100 mm semi-confluent plates of circadian model cell line U2OS:*Bmal1-luc* (*Maier et al., 2009*) were transfected with a set of three Stealth short interfering RNAs targeting *COPS7B* and *COPS4* (25 nM, ThermoFisher Scientific, Waltham, MA). Lipofectamine RNAiMAX transfection reagent was used for efficient siRNAs delivery and the protocol was performed according to the manufacturer's directions (ThermoFisher Scientific). To determine the interaction between COPS7B and clock proteins, 100 mm plates of HEK293T were transfected with FLAG epitope-tagged clock proteins BMAL1, PER2 and CRY1 (5 ug) in the presence or absence of siRNAs targeting *COPS7B* (25 nM). The protocol for Lipofectamine 2000 Transfection Reagent (Invitrogen)-mediated transient transfection was used according to the manufacturer's instructions.

## Protein isolation, Immunoprecipitation, Western Blots and Antibodies

### Preparation of nuclear protein extract

72 hr post-transfection, HEK293T or U2OS cells transiently transfected with FLAG epitope-tagged clock proteins (BMAL1, PER2 and CRY1) were synchronized by dexamethasone as described above, and harvested at the maximal protein levels for BMAL1 (12 hr after dexamethasone shock), PER2 and CRY1 (2 hr after dexamethasone shock) to obtain nuclear protein lysates. Nuclear extracts were prepared according to the method of *Schreiber et al., 1989*.

### Preparation of whole cell protein extract

RIPA lysis buffer (0.02M Tris/pH7.2, 0.15M NaCl, 1% Trion X-100, 1% Na-deoxycholate, 0.1% SDS) was used to prepare whole cell protein extracts. Briefly, transfected HEK293T resp. U2OS cells were washed twice with ice-cold 1xPBS. 300 μl of RIPA lysis buffer supplemented with protease inhibitor cocktail and phosphatase inhibitors (1 mM NAF, 0.1 mM $Na_3VO_4$) was added to the 100 mm plates. After 10 min incubation on ice, cells were scraped vigorously and the suspension was spun down at 10000 RPM at 4°C for 10 min. The supernatant, containing proteins, was carefully transferred into the fresh microfuge tube. The concentration of the proteins was determined using the Pierce BCA Protein assay kit (ThermoFisher Scientific).

### Immunoprecipitation

was performed using standard procedures with the below mentioned adjustments (*Ausubel, 2003*). Nuclear extracts (1000 μg) were bound with antibodies (Monoclonal ANTI-FLAG M2 Antibody/

Sigma, anti COPS7B antibody/Abcam, Cambridge, UK) for 90 min at 4°C, on a horizontal rotator at 10 RPM. The antibody-protein complex was then incubated for 1 hr with protein A agarose beads (Diagenode, Denville, NJ) at 4°C, on a horizontal rotator at 10 RPM. After incubation, the beads were washed once with high-salt buffer (50 mM Tris, 500 mM NaCl, 1% NP-40) followed by two gentle washes in low-salt buffer (50 mM Tris, 120 mM NaCl, 0.5% NP-40). Elution of the proteins from the beads was performed for 3 min at 95°C with 15 µl of 2x SDS sample buffer, containing β-mercaptoethanol. Equal amounts of IP reaction mixtures were loaded on a 9% SDS-PAGE gel together with 1/20 of the IP amounts of input. The protein gel electrophoresis and Western blotting was performed using standard procedures (*Ausubel, 2003*). Equal loading and size detection using a protein ladder were verified by Ponceau-S staining of membranes prior to probing.

## Antibodies

For co-immunoprecipitation, Monoclonal ANTI-FLAG M2 Antibody from Sigma (F3165 RRID:AB_259529) and anti-COPS7b antibody from Abcam (ab124718 RRID:AB_10971678) were used at 1:50 dilution. For detection in co-immunoprecipitation experiments, primary Monoclonal ANTI-FLAG M2 Antibody from Sigma was diluted at 1:1000, primary anti-COPS7b antibody from Abcam was used at 1:1000. For detection of protein levels in an ''around-the-clock'' experiment primary anti-COPS7b antibody from Abcam was used at 1:1000 and Anti-βIII Tubulin mAB from Promega (G7121 RRID: AB_430874) was used at 1:500 dilutions. The probing of the secondary antibody was done at 1:10000 for IRDye 680–goat anti-mouse IgG (926–32220 RRID:AB_621840; Licor, Lincoln, NE) and 1:10000 for IRDye 800–goat anti-rabbit IgG (926–33210 RRID: AB_10796098; Licor).

## Metabolic Pulse-chase

The half-lives of transfected FLAG epitope-tagged circadian clock proteins (BMAL1, PER2 and CRY1) in the absence (as a control) and presence of siRNAs targeting *COPS7B* were measured by metabolic pulse chase protein labeling according to the protocol of Zhou *et al*. Briefly, 72 hr post-transfection, 100 mm plates of HEK293T cells were starved of amino acids for 1 hr and metabolically labeled for 30 min with 100 µCi of Express-35S protein-labeling mix (PerkinElmer Life Sciences, Schwerzenbach, CH) in 1 ml methionine- and cysteine-free DMEM medium supplemented with 10% dialysed FCS ('the pulse'). Sunsequently, cells were washed to remove unbound radioactive amino acids, and incubated in complete medium ('the chase'). At indicated time points (0, 2, 4, and 6 hr), the cells were disrupted in ice-cold RIPA lysis buffer (0.02M Tris/pH7.2, 0.15M NaCl, 1% Trion X-100, 1% Na-deoxycholate, 0.1% SDS), and 1000 µg of whole cell extract were immunoprecipitated with Monoclonal ANTI-FLAG M2 Antibody from Sigma (F3165). Proteins were subjected to 9% SDS-PAGE and the gels were dried for 2 hr at 80°C by Model 583 gel dryer (BIO-RAD, Hercules, CA). Labeled proteins were visualized by autoradiography and quantitative analysis of three independent experiments for each circadian clock protein was performed with ImageStudio software.

## Preparation nuclear protein extracts for SILAC MS analysis

Livers were homogenized in sucrose homogenization buffer containing 2.2 M sucrose, 15 mM KCl, 2 mM EDTA, 10 mM HEPES (pH 7.6), 0.15 mM spermin, 0.5 mM spermidin, 1 mM DTT, and protease inhibitors (0.5 mM PMSF, 10 µg/ml Aprotinin, 0.7 µg/ml Pepstatin A, and 0.7 µg/ml Leupeptin). Lysates were deposited on a sucrose cushion containing 2.05 M sucrose, 10% glycerol, 15 mM KCl, 2 mM EDTA, 10 mM HEPES (pH 7.6), 0.15 mM spermin, 0.5 mM spermidin, 1 mM DTT, and protease inhibitors. After 45 min of centrifugation at 105,000 g at 4°C. The nuclei were suspended in a nucleus buffer composed of 10 mM HEPES (pH 7.6), 100 mM KCl, 0.1 mM EDTA, 10% Glycerol, 0.15 mM spermine, 0.5 mM spermidine, 0.1 mM NaF, 0.1 mM sodium orthovanadate, 0.1 mM ZnSO4, 1 mM DTT, and protease inhibitors (0.5 mM PMSF, 10 µg/ml Aprotinin, 0.7 µg/ml Pepstatin A, and 0.7 µg/ml Leupeptin). Then nuclear protein extracts were obtained by adding an equal volume of NUN buffer (2 M urea, 600 mM NaCl, 50 mM HEPES (pH 7.6), 1 mM DTT, protease inhibitors (cOmplete ULTRA, Roche), a phosphatase inhibitor cocktail (PhosphoSTOP, Roche), and deacetylase inhibitors AGK7, salermide, and trichostatin A (all from SantaCruz Biochemicals, Dallas, TX), followed by a 20 min incubation on ice. The supernatants resulting from a 10 min centrifugation at 21,000 g at 4°C constituted the nuclear extracts (NE) and were used to perform proteins MS-analysis. Protein extracts were quantified using a BCA protein assay kit (Thermo Fisher Scientific).

## Nuclear protein SILAC-MS and data analysis

Mixed samples were reduced with 5 mM DTT and alkylated with 18.75 mM iodoacetamide for 30 min at room temperature in the dark. After an ethanol-acetate precipitation, they were resuspended in 250 mM triethylammonium bicarbonate pH 8.0 containing 4M urea and digested overnight at 37°C with sequencing grade modified trypsin (Promega, Madison, WI) at a 1:50 (w/w) trypsin:protein ratio. The obtained peptide mixtures (250 µg total material) were desalted on SepPak C18 cartridges (Waters Corp., Milford, MA), dried, dissolved in 4M Urea with 0.1% Ampholytes pH 3–10 (GE Health-care) and fractionated by off-gel focusing as described (*Geiser et al., 2011*). The 10 fractions obtained were desalted on a microC18 96-well plate (Waters Corp.), dried and resuspended in 0.1% formic acid, 3% (v/v) acetonitrile for LC-MS/MS analysis. Samples were analyzed on a hybrid linear trap LTQ-Orbitrap Velos mass spectrometer (Thermo Fisher Scientific) interfaced via a nanospray source to a Dionex RSLC 3000 nanoHPLC system (Dionex, Sunnyvale, CA). Peptides were separated on a reversed-phase Acclaim Pepmap nanocolumn (75 µm ID x 25 cm, 2.0 µm, 100 Å, (Dionex)) with a gradient from 5% to 85% acetonitrile in 0.1% formic acid (total time: 200 min) and a flow rate of 300 nl/min. Full MS survey scans were performed at 60'000 resolution. In data-dependent acquisition controlled by Xcalibur 2.1 software (Thermo Fisher Scientific), the twenty most intense multiply charged precursor ions detected in the full MS survey scan were selected for CID fragmentation in the LTQ linear trap with an isolation window of 4.0 m/z and then dynamically excluded from further selection during 35 s.

MS data were analyzed and quantified with MaxQuant version 1.3.0.5 (*Cox and Mann, 2008*), using Andromeda as search software (*Cox et al., 2011*) against UniProt (release 2012_02) database restricted to mouse (*Mus musculus*) taxonomy and a custom database containing usual contaminants (digestion enzymes, keratins, etc). Cleavage specificity was trypsin/P (cleavage after K, R, including KP and RP) with two missed cleavages. Mass tolerances were of 6 ppm for the precursor and 0.5 Da for CID tandem mass spectra. The iodoacetamide derivative of cysteine was specified as a fixed modification, and oxidation of methionine and protein N-terminal acetylation were specified as variable modifications. Protein identifications were filtered at 1% FDR established by MaxQuant against a reversed sequence database. A minimum of one unique peptide was necessary to discriminate sequences which shared peptides. Details of peak quantitation and protein ratio computation by MaxQuant are described elsewhere (*Cox and Mann, 2008*). Raw mass spectrometry data and search engine outputs have been deposited to the ProteomeXchange Consortium (proteomexchange.org) via the PRIDE partner repository (*Vizcaíno et al., 2016*) with the dataset identifier PXD003818.

## Rhythmicity analysis for nuclear proteins

We assessed the rhythmicity in temporal nuclear accumulation of proteins using harmonic regression, as previously (*Mauvoisin et al., 2014*). Briefly, focusing on only 24 hr periodicity (data are generated under 24 hr LD cycles), we used a multiple linear regression for each relative protein time trace $y(t)$ (transformed to log2 units). For this analysis, we used the relation:

$$y(t) = \mu + a\sin\left(2\pi\frac{t}{24}\right) + b\cos\left(2\pi\frac{t}{24}\right) + \text{noise}$$

where $\mu$ is the mean, whereas $a$ and $b$ are the coefficients of cosine and sine functions with period of 24 hr, respectively. The resulting p-values of all proteins are used to estimate false discovery rate (FDR) by the Benjamini–Hochberg method (*Benjamini and Hochberg, 1995*).

## Acknowledgements

LG has been supported by the Swiss National Science Foundation. This work has received further support via SAB from the Swiss National Science Foundation, the Velux Foundation, and the Clinical Research Priority Program 'Sleep and Health' of the University of Zürich. LG and SAB are members of the Zurich Neurozentrum, a division of the Life Sciences Zürich graduate program. We thank Juergen Ripperger and Urs Albrecht for generous donation of antibodies. Research in the FG laboratory received funding from the European Research Council (through individual Starting Grants ERC-2010-StG-260988) and the Leenaards Foundation. KP was supported by the 5 Top 100 Russian Academic Excellence Project at the Immanuel Kant Baltic Federal University.

## Additional information

### Funding

| Funder | Grant reference number | Author |
| --- | --- | --- |
| Schweizerischer Nationalfonds zur Förderung der Wissenschaftlichen Forschung | CRSII3_160741 | Steven A Brown |
| Zurich Hospital | CRPPSleep&Health | Steven A Brown |
| Velux Fonden | 923 | Steven A Brown |
| European Research Council | ERC-2010-StG-260988 | Frederic Gachon |
| Fondation Leenaards | Grant | Frederic Gachon |
| Immanuel Kant Baltic University | 5 Top 100 Russian Academic Excellence Project | Konstantin Popadin |

The funders had no role in study design, data collection and interpretation, or the decision to submit the work for publication.

### Author contributions

Ludmila Gaspar, Conceptualization, Investigation, Methodology, Writing—original draft, Project administration; Cedric Howald, Data curation, Investigation, Writing—original draft; Konstantin Popadin, Data curation, Investigation, Methodology; Bert Maier, Resources, Investigation, Methodology, Writing—original draft; Daniel Mauvoisin, Resources, Data curation, Investigation; Ermanno Moriggi, Conceptualization, Formal analysis, Investigation; Maria Gutierrez-Arcelus, Resources, Investigation, Methodology; Emilie Falconnet, Investigation, Methodology; Christelle Borel, Conceptualization, Resources, Data curation, Writing—review and editing; Dieter Kunz, Conceptualization, Resources, Investigation, Methodology; Achim Kramer, Emmanouil T Dermitzakis, Stylianos E Antonarakis, Conceptualization, Resources, Funding acquisition, Writing—review and editing; Frederic Gachon, Conceptualization, Resources, Funding acquisition, Methodology, Writing—review and editing; Steven A Brown, Conceptualization, Formal analysis, Supervision, Funding acquisition, Project administration, Writing—review and editing

### Author ORCIDs

Ludmila Gaspar, http://orcid.org/0000-0001-6673-7492
Konstantin Popadin, https://orcid.org/0000-0002-2117-6086
Daniel Mauvoisin, https://orcid.org/0000-0003-0571-0741
Ermanno Moriggi, http://orcid.org/0000-0002-4600-5777
Frederic Gachon, https://orcid.org/0000-0002-9279-9707
Steven A Brown, http://orcid.org/0000-0001-5511-568X

### Ethics

Human subjects: All human samples used in these studies were obtained after approval of all protocols and procedures by the relevant responsible authorities (of the University Hospital Geneva, CH; and Charite Universitätsmedezin, Berlin, DE), and prior written informed consent was obtained from all subjects or their legal guardians.
Animal experimentation: All animal experiments were conducted with the approval of relevant cantonal veterinary authorities in Switzerland, after prior review of all procedures and planned experiments.

### Decision letter and Author response

Decision letter https://doi.org/10.7554/eLife.24994.027
Author response https://doi.org/10.7554/eLife.24994.028

# Additional files

## Supplementary files
• Supplementary file 1. Spearman correlations between the sets of positive alleles (p<10$^{-5}$) identified for all phenotypes.
DOI: https://doi.org/10.7554/eLife.24994.021

• Supplementary file 2. List of genes associated with positive SNPs, determined via GWAS, Hi-C and RTC analysis. Right columns, unadjusted p-values and –logP values.
DOI: https://doi.org/10.7554/eLife.24994.022

• Supplementary file 3. List of genes implicated in the pathways determined via Gsea4GWAS analysis. Middle columns, unadjusted p-values and –logP values. Right column, lowest-level GO term associated with each pathway.
DOI: https://doi.org/10.7554/eLife.24994.023

• Transparent reporting form
DOI: https://doi.org/10.7554/eLife.24994.024

## Major datasets
The following previously published dataset was used:

| Author(s) | Year | Dataset title | Dataset URL | Database, license, and accessibility information |
|---|---|---|---|---|
| Dermitzakis E | 2011 | Gencord | https://ega-archive.org/datasets/EGAD00000000027 | Publicly available at the NCBI Sequence Read Archive (accession no. EGAD00000000027) |

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
