## [Decision Letter]

Thank you for submitting your article "The genomic landscape of human cellular circadian variation points to a novel role for the human signalosome" for consideration by *eLife*. Your article has been reviewed by three peer reviewers, one of whom, Patrick Nolan (Reviewer #1), served as Guest Reviewing Editor, and the evaluation has been overseen by Mark McCarthy as the Senior Editor. The following individuals involved in review of your submission have agreed to reveal their identity: Yoshitaka Fukada (Reviewer #2) and Michael McCarthy (Reviewer #3).

The reviewers have discussed the reviews with one another and the Reviewing Editor has drafted this decision to help you prepare a revised submission.

Summary:

Gaspar et al. present their findings on circadian variation in human umbilical cord fibroblasts. They used a GWAS study and identified SNPs associated with a number of molecular circadian phenotypes. The authors claim the genetic variants they identified show greater than expected differences in allele frequency in a group of subjects with extreme chronotypes. Based on GWAS data, they focused on COPS7B, a subunit of the COP9 signalosome, for a number of mechanistic investigations into period determination. The authors use a number of approaches, including RNAi, Co-IP of FLAG-tagged proteins and SILAC, to validate their initial findings and propose some interesting and novel molecular mechanisms. Specifically, they argue that COPS7B may promote the stability of the circadian transcriptional regulator, BMAL1. Based on these observations, they posit that clock protein half-life is determined by contributions both from destabilising factors via the ubiquitin proteasome and stabilising factors via the signalosome. The latter would represent a significant addition to our understanding of the molecular basis of circadian period determination.

Essential revisions:

GWAS:

Conducting a GWAS study with such a small number of samples (140-147) is ambitious and this probably underlies the high p-values for the data described throughout the manuscript. Although such datasets usually require higher power, the reviewers understand the limitations here. It is not reasonable to expect numbers in a cell-based experiment to go much larger, or equal to those seen in GWAS (N>10,000). Both points should be discussed and justified.

Please justify multiple testing using a threshold of FDR 0.1. Why choose this threshold? An FDR of q<0.05 is reasonable for a GWAS study, even a cell based study. An FDR of 0.1 seems unusually high. If not justifiable, then perhaps gene list should be modified.

In subsequent text, the authors quote the p value for the two most significant SNPs affecting PER2 expression, not the q value. Why?

It would be helpful to know the degree of phenotypic correlation across the 5 rhythm parameters, and to estimate their genetic overlap. Do any of the markers associate with more than one parameter (as expected)? Or if not, why? For example, in comparing Manhattan plots, PER2 expression is the only variable where the Chr2 SNPs show anything near a threshold. Given the eventual mechanistic validation (BMAL1 stability), shouldn't we expect a similar pattern in the Manhattan plot for PER1 expression? Please provide this information and justify.

In Figure 1, the statistical method is not well described, nor is a citation provided that validates the approach. The reviewers have concerns that simply calculating the difference in allele frequency between groups may not be valid across the entire range of allele frequencies. Why wasn't a more standard approach employed (e.g. chi square analysis, or for selected SNPs, trait morningness vs. genotype association)? If possible, these analyses should be included. Also in Figure 1, is it appropriate to study all association with chronotype equally? For instance, SNP associations with phase may be predicted to have stronger contribution to chronotype than amplitude. This approach should be justified, especially in light of the small sample size and the multiple comparisons problem.

RNAi:

Aspects of the RNAi experiments are hard to interpret. Of the 28 genes affecting period and 29 affecting amplitude, is there overlap (i.e. genes that affect both)? Of the control RNAis, the results should more clearly state how many control RNAis were used, and what proportion affected period or amplitude. Does a chi-square test indicate this proportion is significantly different?

In subsection “Genes associated with positive SNPs affect circadian clock function”, when assessing the probability that the genes affect the circadian clock when compared to random genesets, it is unclear why the authors have used one test to assess the reduced & increased period/amplitude and a different test for the combined cumulative distribution. Presumably this is from analysis of the same data so the same tests should be used, please change or justify.

Gene enrichment analysis, Figure 3:

The authors state that they 'applied gene ontology enrichment analysis to the list that we obtained'. There is a bit of confusion here. Was this applied to the 59 genes from 3C or the 76 genes from the original eQTL? What threshold was used for the GSEA? Figure 3 should be amalgamated into a graph with clear p-values added to the graph. As written in subsection “Positive SNPs highlight protein catabolism as essential to human circadian variation”, paragraph 2, authors then go on to pick the genes from 5 of the top 6 pathways for further analysis shown in Figure 3. Is this somehow relevant to what is shown in Figure 3? Only some of the pathways shown in Figure 3 are significant and included in the top 6 pathways. For example, oxidoreductase activity (included) is not in the top 6 pathways in Figure 3—figure supplement 1 while macromolecule catabolic process (not included) is? Finally, given that the resulting genes are not significant in the GWAS and are not from a random sample, please provide evidence that the significant increase/decrease in period (Figure 3) is not by chance.

Figure 4 and associated text:

Is the rs920400 AA genotype associated with long or short period in fibroblasts? How can this be reconciled with 'average PER2 expression' being higher? Is it justifiable to use a single timepoint here to look at PER2 expression? Do the genetic association findings of COPS7B share any association with the BMAL-Luc parameters? As the main result, the COPS7B association should be reported for all of the 5 parameters selected. If the genetic association does not match the RNAi experiment, the discrepancy should be discussed. It is difficult to accept that COPS4 knockdown lengthens period (by 9.9 hrs!) as it appears to completely obliterate the bioluminescence rhythm. What is the condition of the cells after knockdown? Labelling in Figure (COP4) is incorrect.

Figure 5 and associated text:

The authors claim direct binding of COP9 signalosome with several clock proteins, but the data need to be polished in order to fully support this conclusion. The immunoprecipitation analysis (Figure 5) lacks essential experimental controls: in the FLAG-IP experiment, COPS7b signal should be no longer be detected in the absence of co-expression of FLAG-proteins in the same gel. These controls should be included in Western Blots. In the FLAG IP (a) the COPS7B antibody appears to cross-react with both PER2-FLAG and CRY1-FLAG. This confound should be addressed and justified. Both panels seem to show that interaction of COPS7B with PER2-FLAG and CRY1-FLAG is greater than with BMAL1-FLAG, yet the authors are proposing a direct interaction with BMAL1, how is this so? Authors should address this as it affects their arguments in the paper. Figure 5. What does the * signify and how was it measured? Presumably this should be a repeated measures ANOVA determination. Statistical methods should be included. How does this translate into 'the half-life of BMAL1 was reduced twofold'? Figure 5—figure supplement 1. Shouldn't you be seeing some protein degradation in these graphs, there is nothing evident here, particularly so in the CRY1 experiment. Some explanation is required in the text. How is this related to 'The half lives of CRY1 and PER2 remained unchanged'? Neither levels approach anything near 50% of the time-0 timepoint. What does 'even if their absolute abundance varied' mean?

Subsection “COPS7B and the COP9 signalosome are rhythmically imported into the nucleus”:

The argument the authors make about the conflicting phases of expression of PER/CRY with COPS7B in U2OS cells might hold for native protein interactions but this should not hold when constitutively ovexpressing recombinant PER-FLAG and CRY-FLAG proteins. There must be some other cause for the discrepancy seen in protein interactions in the two cell lines. Please address these issues in Results and Discussion.

Presentation and style:

There are numerous instances of errors in cross-referencing of Figures, referencing citations and overall flow of text in Results section.

Abstract:

Given that the Abstract is the first port-of-call for many readers, it should be a realistic reflection of the main findings and conclusions in the manuscript. Some text such as 'Overwhelmingly gene set enrichment points to differences in protein catabolism as the major source of clock variation in humans' is rather strong. The reviewers request that this be toned down appropriately.

---

## [Author Response]

Essential revisions:GWAS:Conducting a GWAS study with such a small number of samples (140-147) is ambitious and this probably underlies the high p-values for the data described throughout the manuscript. Although such datasets usually require higher power, the reviewers understand the limitations here. It is not reasonable to expect numbers in a cell-based experiment to go much larger, or equal to those seen in GWAS (N>10,000). Both points should be discussed and justified.

These are indeed a valid point, and we thank the reviewers for their understanding of the problems of cell-based genetics. (Even as it stands, ours was a story close to a decade in the making.) We have now discussed the points raised by this reviewer:

“Typically, because of the labor involved in collecting and analyzing cellular material, the cohorts used in such investigations are much smaller (hundreds of samples rather than >10,000 in conventional GWAS). Partially compensating for this deficiency is the relative simplicity of the analytical system and the precision of expression trait measurement.”

and again:

“While multiple suggestive alleles were identified, it should be noted that none achieved “genome wide” significance, as might be expected from a cell-based study of moderate sample size and 2.5million SNPs tested. Therefore, we undertook further experiments to demonstrate the relevance of our results.”

Please justify multiple testing using a threshold of FDR 0.1. Why choose this threshold? An FDR of q<0.05 is reasonable for a GWAS study, even a cell based study. An FDR of 0.1 seems unusually high. If not justifiable, then perhaps gene list should be modified.

Given the multistep selection process we used, we preferred to tolerate a larger rate of false positives initially (1/10 vs. 1/20) that would be screened out later. It should be noted that “suggestive” significance for GWAS is generally defined at 10e-5, and this corresponds in our study to an FDR of 0.1. It should further be noted that our FDR is based upon a strict correction for the 2.5million SNPs tested, and for European genomes the 2.5M SNP chip that we used is in fact “far too dense”: it is estimated that only 260,000 “tag-SNPs” are sufficient to capture variance in all common SNPs in the Phase I Hapmap project for European genomes. Thus, a Bonferroni-based correction is in reality 10x too severe! An excellent discussion of this point can be found here: Nature. 2005 Oct 27; 437(7063): 1299– 1320.

Finally, the validation of this strategy is in our opinion visible by the success of the GSEA approach, which demonstrated the functionally related significance of these hits.

We now discuss our rationale explicitly in the text:

“We chose this threshold to correspond to “suggestive” significance (uncorrected p<1x10^-5^) in genome-wide association: since it is estimated that only a tenth of the SNPs on the high-density chip that we used would be necessary to capture all common variation in individuals of European origin, a corrected FDR is therefore up to tenfold “over-corrected”. We therefore preferred to select genes at a relaxed FDR, and subsequently apply further criteria.”

In subsequent text, the authors quote the p value for the two most significant SNPs affecting PER2 expression, not the q value. Why?

For the “raw” GWAS, we wanted the values to be immediately comparable across other genetic studies, and p values have been reported much longer than q values. The main reason for the persistence in the use of p-values in GWAS is that correction for multiple testing after genotyping with high-density arrays is highly disputable. Essentially, the 2.5million SNPs on a modern chip are not independent. In fact, the Hapmap project shows that the oversampling rate is tenfold for Europeans. By leaving these values as p-values, anyone can easily correct however they want.

It would be helpful to know the degree of phenotypic correlation across the 5 rhythm parameters, and to estimate their genetic overlap. Do any of the markers associate with more than one parameter (as expected)? Or if not, why? For example, in comparing Manhattan plots, PER2 expression is the only variable where the Chr2 SNPs show anything near a threshold. Given the eventual mechanistic validation (BMAL1 stability), shouldn't we expect a similar pattern in the Manhattan plot for PER1 expression? Please provide this information and justify.

We now include the Spearman correlation pairwise across all phenotypes as a way of estimating genetic overlap. We observe varying degrees of global correlation across parameters – i.e. a nonrandom association of p values across the different phenotypes assayed. As might be expected, in particular *PER2* expression correlates with *PER1* expression, and period correlates with phase (a new Table 1). However, this reviewer correctly notes that correlation across top hits is surprisingly absent – i.e. wouldn’t a top hit for *PER2* expression also be one for *PER1* or period length? Apparently not.

We now note this explicitly in the text:

“We next compared SNPs across the traits that we identified. Considering only alleles achieving “suggestive” genome-wide significance (p<10^-5^), we see correlation between multiple pairs of traits, notably PER1 and PER2 expression, and period and phase (Table 1). Given the close homology and overlapping function of the PER proteins, and the known correlation between period length and phase, these correlations are expected. Surprisingly, however, the most significant alleles in any one category are not among the most significant alleles in another, suggesting relatively independent genetic regulation of key circadian clock parameters (“state variables”) for small changes, and the resilience of the overall circadian mechanism to small perturbations.”

We continue the theme in our Discussion:

“It is worth noting that although knockdown of COP9 subunits altered circadian period, phase, and PER2 expression, the COPS7B-associated alleles rs920400 and rs10195385 that we genetically identified were associated significantly only with PER2 expression in vitro. More generally, although there was detectable and significant genotypic correlation between period and phase, and PER1 and PER2 expression in our screen, global genetic correlation among these different phenotypes was surprisingly weak. We attribute these differences to the idea that small changes in cellular clock properties can be robustly tolerated without affecting other clock parameters. Such an idea has also been supported by cellular dosage studies individual clock proteins.”

In Figure 1, the statistical method is not well described, nor is a citation provided that validates the approach.

We thought long and hard about how to demonstrate an enrichment that would be valid in a moderately-sized pool of only 40 individuals of extreme chronotype, and invented this method for the task. The method is analogous to those that verify individual eQTL hits in biochemical pathways by validating unequal allele penetrance in individuals with diseases affecting that pathway (e.g. Loeuillet et al., 2008).

We now describe it in better detail in a new section of the Materials and methods.

The reviewers have concerns that simply calculating the difference in allele frequency between groups may not be valid across the entire range of allele frequencies.

This is an important issue that we considered carefully – e.g. an extremely rare allele, present in only one subject who was an extreme early or late type, would skew the results at stringent p values where relatively few alleles were present. We combated this in multiple ways: first, our study already removed rare alleles (minor allele <5%) from consideration. Secondly, we included these data in two different versions: one with all SNPs, and one with common tag-SNPs (Figure 1 and Figure 1—figure supplement 4). This corrects for the multiple comparison problem – the tag-SNP approach counts all alleles in a hapblock only once – and also by definition eliminates very rare alleles). Finally, to fully satisfy this concern, we have gone back and looked specifically at all SNPs p<10-6. The rarest minor allele among all these SNPs is 0.1524. Therefore, we believe that this concern, though valid, is not a problem in our analysis.

Why wasn't a more standard approach employed (e.g. chi square analysis, or for selected SNPs, trait morningness vs. genotype association)? If possible, these analyses should be included.

We respectfully disagree with this idea. Because of the limited number of extreme chronotypes in our pool, we do not think that it is fair to pick single alleles and show their enrichment. This is something that would be easy to do, but we do not think it would be valid. The whole strength of the method we devised is that it avoids singling out any particular allele, and shows that on the whole alleles considered at more stringent p values have greater influence upon chronotype than those at lesser.

Also in Figure 1, is it appropriate to study all association with chronotype equally? For instance, SNP associations with phase may be predicted to have stronger contribution to chronotype than amplitude. This approach should be justified, especially in light of the small sample size and the multiple comparisons problem.

Given the limited number of alleles we obtain, we think that it is important to consider them collectively in order to add power to subsequent pathway analyses. Moreover, while it is commonly predicted that some clock properties affect chronotype more than others, this relationship is far from clear in the literature. For example, we ourselves have shown that amplitude directly affects entrained phase, because a clock of high amplitude will show a smaller phase shift response. Thus, humans with high amplitude tend to be later chronotypes (Brown, PNAS 2008).

RNAi:Aspects of the RNAi experiments are hard to interpret. Of the 28 genes affecting period and 29 affecting amplitude, is there overlap (i.e. genes that affect both)?

At the biochemical level, absolutely – COP9 subunits are one particularly relevant example. At the genetic level, mostly not – i.e., as already mentioned in the context of Figure 1, we do not find highly significant polymorphisms affecting both parameters. However, we do not find this so surprising: in vitro, black-and- white changes in one circadian parameter definitely affect another. However, with smaller changes, these parameters behave much more independently. A good example of this can be seen in the dose-dependent RNAi against clock genes seen in Baggs et al., 2009.

Of the control RNAis, the results should more clearly state how many control RNAis were used, and what proportion affected period or amplitude. Does a chi-square test indicate this proportion is significantly different?

We believe that a misunderstanding occurred here, and we have therefore rewritten our description of this experiment in subsection “Positive SNPs highlight protein catabolism as essential to human circadian variation” and in the corresponding figure legends. Although only a subset of randomly chosen control RNAis were shown as discrete black circles, the underlying curves represented the entire genome, and the statistical tests described were against this control. However, it is impossible to plot aesthetically so many individual values. Statistical tests for all comparisons except one are significant (as noted in the text), and all exact p-values are cited in the text.

In subsection “Genes associated with positive SNPs affect circadian clock function”, when assessing the probability that the genes affect the circadian clock when compared to random genesets, it is unclear why the authors have used one test to assess the reduced & increased period/amplitude and a different test for the combined cumulative distribution. Presumably this is from analysis of the same data so the same tests should be used, please change or justify.

We have now homogenized our presentation here (Figure 2 and Figure 3). In fact, both types of analyses were already present in both comparisons (against the whole genome, not random genesets as mentioned above), but one was in the main figure and the other in supplements. We chose two different representations so that the reader could see in one case the change in magnitude of phenotype, and in the other case a directional shift in the distribution. That is also why two different statistical tests were cited. We think that this approach is valid, but in any case both approaches and both tests were used in both cases, and now the figures look homogenous.

Gene enrichment analysis, Figure 3:The authors state that they 'applied gene ontology enrichment analysis to the list that we obtained'. There is a bit of confusion here. Was this applied to the 59 genes from 3C or the 76 genes from the original eQTL?

Figure 3 were generated from the original list obtained purely by individual gene q-value. 3c-f were a completely separate analysis done using the GSEA algorithm, and we highlight the similarity of the results in the text. Thus, these are two completely separate analyses using different methods that nevertheless obtained similar results.

We have added additional phrases to make this distinction clearer.

“We first applied gene ontology enrichment analysis to the list of 60 genes…”

What threshold was used for the GSEA?

GSEA intentionally uses a highly relaxed stringency for individual alleles (10e-4) and then specifically looks for enriched sets. The thresholds we have used here are consistent with those used previously, and recommended by the authors of the tool. We have now noted this relaxed stringency explicitly in the subsection “COPS7B influences PER2 expression and clock function”.

Figure 3 should be amalgamated into a graph with clear p-values added to the graph.

We have now exchanged the pathway diagram of Figure 3 for a simplified graph, and added as supplements the full pathway diagram, as well as a supplementary file with IDs for the GO biological processes, the genes from our gene set within the pathways, and the adjusted p-values. Figure 3, the GOslim functional distribution, represents complementary information. It shows the highest-level functional annotations of the groups of genes identified. Since it is showing the actual annotation rather than the probability of mapping by chance, a p value is not appropriate, and indeed is not supplied with the tool.

As written in subsection “Positive SNPs highlight protein catabolism as essential to human circadian variation”, paragraph 2, authors then go on to pick the genes from 5 of the top 6 pathways for further analysis shown in Figure 3. Is this somehow relevant to what is shown in Figure 3? Only some of the pathways shown in Figure 3 are significant and included in the top 6 pathways. For example, oxidoreductase activity (included) is not in the top 6 pathways in Figure 3—figure supplement 1 while macromolecule catabolic process (not included) is?

Clearly, this was not very well explained. The raw GSEA output is shown in Figure 3—figure supplement 2. Five of the top six pathways were different aspects of protein catabolism, and this is what we chose to investigate in the rest of the paper. Thus, the following panels d-f, and the remainder of the paper, deal only with protein stability. However, several other pathways were identified at p<0.003 or better, and we listed each only once in the main Figure 3. We now explain this abbreviation better in the text:

“The full list from GSEA analysis is shown in Figure 3—figure supplement 2. We also show an abbreviated list, eliminating major redundancies in pathway annotations, in Figure 3. Since by far the most significant category identified was protein catabolism, we chose to explore further the significance of alleles in this group.”

Finally, given that the resulting genes are not significant in the GWAS and are not from a random sample, please provide evidence that the significant increase/decrease in period (Figure 3) is not by chance.

For this, we used a Kolmogorov-Smirnov test to show that the distributions are different. The comparison is against the genome as a whole, and the KS test p- value already accounts for differences in the number of elements in the two distributions.

Figure 4 and associated text:Is the rs920400 AA genotype associated with long or short period in fibroblasts? How can this be reconciled with 'average PER2 expression' being higher?

Short period. There is an obvious contradiction here that we already discuss in the first paragraph of the Discussion. Deletion studies by multiple other labs show that PER protein knockdown is associated with a short period. We think that the increase in *Per2* RNA levels is in fact an indirect effect due to an increase in BMAL1 protein.

Is it justifiable to use a single timepoint here to look at PER2 expression?

While many timepoints might be useful for a circadian question, it was not feasible within the context of the hundreds of samples we examined. Since we attribute the transcriptional phenotype to an indirect effect, we decided not to pursue *Per2* expression in further detail. Clearly, altered timing of expression would be reflected in changed levels at a single timepoint. However, our main goal here was to verify the findings of the screen itself.

We have added the following text so that the reader is aware of this potential confound:

“It should be noted that here and in the screen itself, a single timepoint was used to investigate PER2 expression, corresponding to its expected first peak after cell line synchronization. Thus for any single subject, absolute changes in level could also reflect changes in timing. Therefore, we next explored this effect using exogenous assays.”

Do the genetic association findings of COPS7B share any association with the BMAL-Luc parameters? As the main result, the COPS7B association should be reported for all of the 5 parameters selected. If the genetic association does not match the RNAi experiment, the discrepancy should be discussed.

The only striking association is with *PER2* expression. We now report all associations as requested. Like this reviewer, I would have preferred that a striking association with *PER2* expression would be associated with an equally striking change in period. This is not the case. As requested, we now discuss this explicitly in the text:

“It is worth noting that although knockdown of COP9 subunits altered circadian period, phase, and PER2 expression, the COPS7B-associated alleles rs920400 and rs10195385 that we genetically identified were associated significantly only with PER2 expression in vitro. More generally, although there was detectable and significant genotypic correlation between period and phase, and PER1 and PER2 expression in our screen, global genetic correlation among these different phenotypes was surprisingly weak. We attribute these differences to the idea that small changes in cellular clock properties can be robustly tolerated without affecting other clock parameters. Such an idea has also been supported by cellular dosage studies individual clock proteins.”

It is difficult to accept that COPS4 knockdown lengthens period (by 9.9 hrs!) as it appears to completely obliterate the bioluminescence rhythm. What is the condition of the cells after knockdown? Labelling in Figure (COP4) is incorrect.

This reviewer refers to the fact that cellular toxicity lengthens circadian period. However, these cells were viable and continued to grow, but are basically arrhythmic. The reduction in bioluminescence was not due to cell death. We have highlighted this effect better by now including raw bioluminescence traces.

Figure 5 and associated text:The authors claim direct binding of COP9 signalosome with several clock proteins, but the data need to be polished in order to fully support this conclusion. The immunoprecipitation analysis (Figure 5) lacks essential experimental controls: in the FLAG-IP experiment, COPS7b signal should be no longer be detected in the absence of co-expression of FLAG-proteins in the same gel. These controls should be included in Western Blots.

Since we did the immunoprecipitation in both directions, we initially judged this control redundant. However, we have now done this control as suggested by the reviewer and we see no false positive signal. The results from this experiment are now included in Figure 5—figure supplement 1.

In the FLAG IP (a) the COPS7B antibody appears to cross-react with both PER2-FLAG and CRY1-FLAG. This confound should be addressed and justified.

This is not the case, but is rather an unfortunate background of slightly different size. Short of trimming the bands more closely, we can’t make this problem “disappear”. We now make reference to it explicitly in the legend.

Both panels seem to show that interaction of COPS7B with PER2-FLAG and CRY1-FLAG is greater than with BMAL1-FLAG, yet the authors are proposing a direct interaction with BMAL1, how is this so? Authors should address this as it affects their arguments in the paper.

This is an important point that we address in Discussion paragraph three, and have now expanded here and in paragraph four. In transfection experiments into noncircadian cells, we do indeed see interaction with all three-clock proteins mentioned. In circadian cells, we see only the BMAL1 interaction, and subsequently show that the COPS9 signalosome is imported into the nucleus roughly coincident with BMAL1 in cells and tissues. Thus, whether or not it could interact with all three, functionally it appears to interact primarily with BMAL1.

Figure 5. What does the * signify and how was it measured? Presumably this should be a repeated measures ANOVA determination. Statistical methods should be included.

We were simpler here: this was only a Student T test for the replicates at each timepoint, p=0.049. We now describe it in the legend.

How does this translate into 'the half-life of BMAL1 was reduced twofold'? Figure 5—figure supplement 1. Shouldn't you be seeing some protein degradation in these graphs, there is nothing evident here, particularly so in the CRY1 experiment. Some explanation is required in the text. How is this related to 'The half lives of CRY1 and PER2 remained unchanged'? Neither levels approach anything near 50% of the time-0 timepoint. What does 'even if their absolute abundance varied' mean?

We thank the reviewer for pointing out this problem with our quantification. As the blots themselves show, degradation is indeed occurring, and it is not different between wildtype and knockdown. Values approach 50% at 2-4 hours. Curiously, though, they subsequently “flatten” during the 6 hours analysed; these experiments were repeated multiple times. While we do not understand why further degradation does not occur – this might even be an interesting regulatory phenomenon – we prefer not to speculate, and simply use the assay to calculate differences between wildtype and knockdown.

Subsection “COPS7B and the COP9 signalosome are rhythmically imported into the nucleus”:The argument the authors make about the conflicting phases of expression of PER/CRY with COPS7B in U2OS cells might hold for native protein interactions but this should not hold when constitutively ovexpressing recombinant PER-FLAG and CRY-FLAG proteins. There must be some other cause for the discrepancy seen in protein interactions in the two cell lines. Please address these issues in Results and Discussion.

We have now qualified our conclusions as suggested by this reviewer:

“It should be noted that this result does not completely explain why interactions with other clock proteins would not be observed when they are exogenously transfected, and suggests therefore that time-specific cofactors might be required for the interaction.”

…and then:

“Although we could detect interactions with multiple transfected clock proteins in nonrhythmic HEK cells, in rhythmic U2OS cells only BMAL1 interaction was detectable, suggesting further that other time-of-day-specific factors might be required for this interaction.”

Presentation and style:There are numerous instances of errors in cross-referencing of Figures, referencing citations and overall flow of text in Results section.

We apologise for these errors. Most result from a reformatting for *eLife*, followed by accidental download of the wrong version. Still, we should have noticed. We believe that all these errors have now been corrected.

Abstract:Given that the Abstract is the first port-of-call for many readers, it should be a realistic reflection of the main findings and conclusions in the manuscript. Some text such as 'Overwhelmingly gene set enrichment points to differences in protein catabolism as the major source of clock variation in humans' is rather strong. The reviewers request that this be toned down appropriately.

We have weakened the wording to: “Gene set enrichment points to differences in protein catabolism as one major source of clock variation in humans.”